# Diagnostic implications of pitfalls in causal variant identification based on 4577 molecularly characterized families

Lama AlAbdi[1,2], Sateesh Maddirevula[2], Hanan E. Shamseldin[2], Ebtissal Khouj[2], Rana Helaby[2], Halima Hamid ®[1,2], Aisha Almulhim[1,2], Mais O. Hashem[2], Firdous Abdulwahab[2], Omar Abouyousef[2], Mashael Alqahtani[2], Norah Altuwaijri[3], Amal Jaafar[2], Tarfa Alshidi[2], Fatema Alzahrani[2], Mendeliome Group* & Fowzan S. Alkuraya ®[2,4] ✉

Despite large sequencing and data sharing efforts, previously characterized pathogenic variants only account for a fraction of Mendelian disease patients, which highlights the need for accurate identification and interpretation of novel variants. In a large Mendelian cohort of 4577 molecularly characterized families, numerous scenarios in which variant identification and interpretation can be challenging are encountered. We describe categories of challenges that cover the phenotype (e.g. novel allelic disorders), pedigree structure (e.g. imprinting disorders masquerading as autosomal recessive phenotypes), positional mapping (e.g. double recombination events abrogating candidate autozygous intervals), gene (e.g. novel gene-disease assertion) and variant (e.g. complex compound inheritance). Overall, we estimate a probability of 34.3% for encountering at least one of these challenges. Importantly, our data show that by only addressing non-sequencing-based challenges, around 71% increase in the diagnostic yield can be expected. Indeed, by applying these lessons to a cohort of 314 cases with negative clinical exome or genome reports, we could identify the likely causal variant in 54.5%. Our work highlights the need to have a thorough approach to undiagnosed diseases by considering a wide range of challenges rather than a narrow focus on sequencing technologies. It is hoped that by sharing this experience, the yield of undiagnosed disease programs globally can be improved.

Mendelian diseases span all body systems and developmental stages. Thousands of genes have already been linked to these diseases and thousands more will likely be linked in the future. Until recently, molecular diagnosis of Mendelian diseases relied on their successful clinical delineation such that one or a few relevant genes are sequenced clinically. However, next generation sequencing made it possible to overcome this limitation by enabling the sequencing of the entire genome or relevant parts thereof. The clinical utility of causal variant identification in Mendelian diseases entails the provision of a precise diagnostic label, informing management decisions, and

[1]Department of Zoology, College of Science, King Saud University, Riyadh, Saudi Arabia. [2]Department of Translational Genomics, Center for Genomic Medicine, King Faisal Specialist Hospital and Research Center, Riyadh, Saudi Arabia. [3]Department of Clinical Genomics, Center for Genomic Medicine, King Faisal Specialist Hospital and Research Center, Riyadh, Saudi Arabia. [4]Department of Pediatrics, Prince Sultan Military Medical City, Riyadh, Saudi Arabia. *A list of authors and their affiliations appears at the end of the paper. ✉e-mail: falkuraya@kfshrc.edu.sa

empowering individuals (patients and unaffected carriers) to make reproductive choices as well as to understand disease risk in family members and future generations. This underpins the desire to maximize the availability of these tools to end the lengthy diagnostic odyssey and ensure that patients and families achieve their right to an accurate and timely diagnosis[1].

Despite the remarkable advances in Mendelian disease genetics, current technology fails to identify the underlying causal variant in a significant fraction of patients. Large diagnostic exome sequencing (ES) cohorts typically report <50% diagnostic rate[2]. Even whole-genome sequencing (WGS) falls short of attaining the much-anticipated full capture of all Mendelian variants. A recent real-world study on the clinical implementation of WGS in the diagnosis of suspected Mendelian diseases reported a 35% diagnostic rate[3]. It is clear, therefore, that there are persistent challenges in the diagnosis of Mendelian diseases beyond the coverage issue. The identification of these factors will require large scale deep analysis of Mendelian diseases and sharing of results to facilitate the development of robust tools that learn from these pitfalls.

Efforts to characterize the challenges in Mendelian variant identification have been limited and tend to deal with a single challenge at a time e.g. cryptic transcript-deleterious variants[4,5]. The Undiagnosed Diseases Network in the US recently published its experience involving 791 evaluated individuals: 231 received 240 diagnoses, including 35% that were "straightforward"[6]. The very nature of UDN makes it enriched for challenging clinical scenarios so their finding of 90 diagnoses that occurred after prior nondiagnostic exome sequencing and 45 diagnoses that are non-genetic may not be reflective of the overall landscape of Mendelian diseases. A recent overview by the Centers for Genomic Medicine only very briefly listed some of the pitfalls of standard analysis[7]. The examples listed by the authors include WGS to identify a homozygous inversion in *QDPR* missed by exome, RNA-seq to identify an intronic variant in trans with a missense in *DES*, bisulfite sequencing to identify aberrant hypermethylation associated with a pathogenic repeat expansion in the *XYLT1* promoter region, and long-read sequencing to identify an inverted triplication flanked by duplications in a proband with Temple syndrome[7]. Thus, there remains a need for a detailed analysis of a large and unbiased Mendelian cohort to both quantitively and qualitatively describe the encountered pitfalls and inform similar efforts.

Here, we describe the challenges encountered in a large Mendelian genomics program involving 4577 molecularly characterized families. We identify categories of challenges that cover the phenotype, pedigree structure, positional mapping, gene, and variant, and quantify their relative contribution. Our results can inform current and future efforts to improve the diagnostic yield of Mendelian diseases globally.

## Results

### Representativeness of the study cohort

Our cohort comprised 4577 families in which a likely causal variant was identified (out of 8024 families in total). The total number of these variants is 2681 (2131 recessive, 455 dominant, 88 X-linked, 6 Y-linked, and one mitochondrial) and the total number of implicated genes is 1604 (400 lacked OMIM listing of the gene-phenotype assertion at the time of analysis). The overwhelming majority of the included cases came through the research lab (94.5%, 4324 / 4577), while 5.5% came through the clinical lab. Similarly, the overwhelming majority of cases came from Saudi Arabia (~96%) with the remaining ~4% coming as international referrals to our program. Consanguinity (defined as parental relatedness equivalent to third cousin or closer) was documented in 81% and lack of consanguinity was documented in 10.5% (consanguinity was unknown in 8.5% families). There was a broad coverage of disease pathologies typical of large Mendelian genomics programs including neurodevelopmental, dysmorphic/congenital

malformation syndromes, inborn errors of metabolism, hematological, immunological, ophthalmological, audiological, pulmonary, gastrointestinal, connective tissue-related, cardiovascular, skeletal, reproductive, and renal. The age distribution was also broad ranging from the zygote stage to 80 years of age. Our cohort consisted of an almost equal distribution of sex (51.8% males and 47.1% females) while the remaining 1.1% were cases of undetermined sex (typically fetuses).

### Genetic diagnostic challenges

We identified 1570 families (34.3%) in which one or more of the following challenges was observed (Fig. 1 and Supplementary Data 1). The classification of the variants was pathogenic or likely pathogenic in the majority of the variants identified, while the remaining 15% were variants of uncertain significance (based on known genes since variants in novel genes are automatically classified as variants of uncertain significance) (Supplementary Data 1). The variants spanned 861 genes, including candidate genes as well as novel allelic disorders in known morbid genes, the majority of which (87.7%, 221 / 252) achieved at least a moderate level gene-disease assertion (Supplementary Data 1). All variants are submitted to ClinVar and all novel gene-disease assertions to GenCC.

1- Phenotype-related:

i. Phenotypic heterogeneity: Supplementary Data 2 lists the families where the phenotype was sufficiently heterogeneous (intrafamilial or interfamilial) to complicate the original molecular diagnosis (~3% of families). For example, the identification of the causal variant of cleft lip and palate *IRF6*:NM_006147.4:c.179 G > C;p.(Trp60Ser), heterozygous in family F750 was challenging because the phenotype varied widely between frank cleft lip and palate to lip pits that were not always apparent clinically due to the use of cosmetic fillers. On the other hand, inter-familial phenotypic heterogeneity significantly delayed the identification of some founder variants shared by families e.g. we identified the same pathogenic founder *INSR* variant NM_000208.4:c.433 C > T;p.(Arg145Cys) in families where the phenotype ranged from classical hyperinsulinism to asymptomatic.

ii. Phenotypic expansion: in 79 families (5%), the phenotype provided by the referring physician was sufficiently different from the typical phenotypic expression of the implicated gene to make the molecular diagnosis challenging (see Supplementary Data 3). As well, the causal variant in *CDK10* was not considered initially in the interpretation of F5780, a simplex case recruited due to hydrocephalus because this is an atypical presentation of Al Kaissi syndrome (one case of Al Kaissi-related hydrocephalus was published after the initial submission[8]). Another example is family F7829 where one affected fetus was found to have bilateral renal agenesis and a homozygous LOF variant in *CD151* (Fig. 2d), which is typically linked to nephropathy rather than renal agenesis (one report of *CD151*-related renal agenesis appeared after the initial submission[9]). Dual molecular diagnosis was specifically investigated and excluded in families under this category.

iii. Allelism: in 83 families (5.3%), the phenotype is sufficiently different from the one described in the literature that it justifies labeling as a distinct allelic disorder (52 allelic disorders in total). Indeed, the phenotype was considered a distinct allelic disorder that later acquired a distinct OMIM entry in 37 families or remains a candidate for OMIM listing in 46 families. Supplementary Data 4 lists these cases, which include 19 unpublished (Table 1). Interestingly, most instances of allelism can be attributed to the recessive nature of the identified variant compared to the dominant ones in the literature and these will be discussed later under gene-specific challenges. Several exceptions are worth noting: Family F8629 presented with primary amenorrhea and infertility and was found

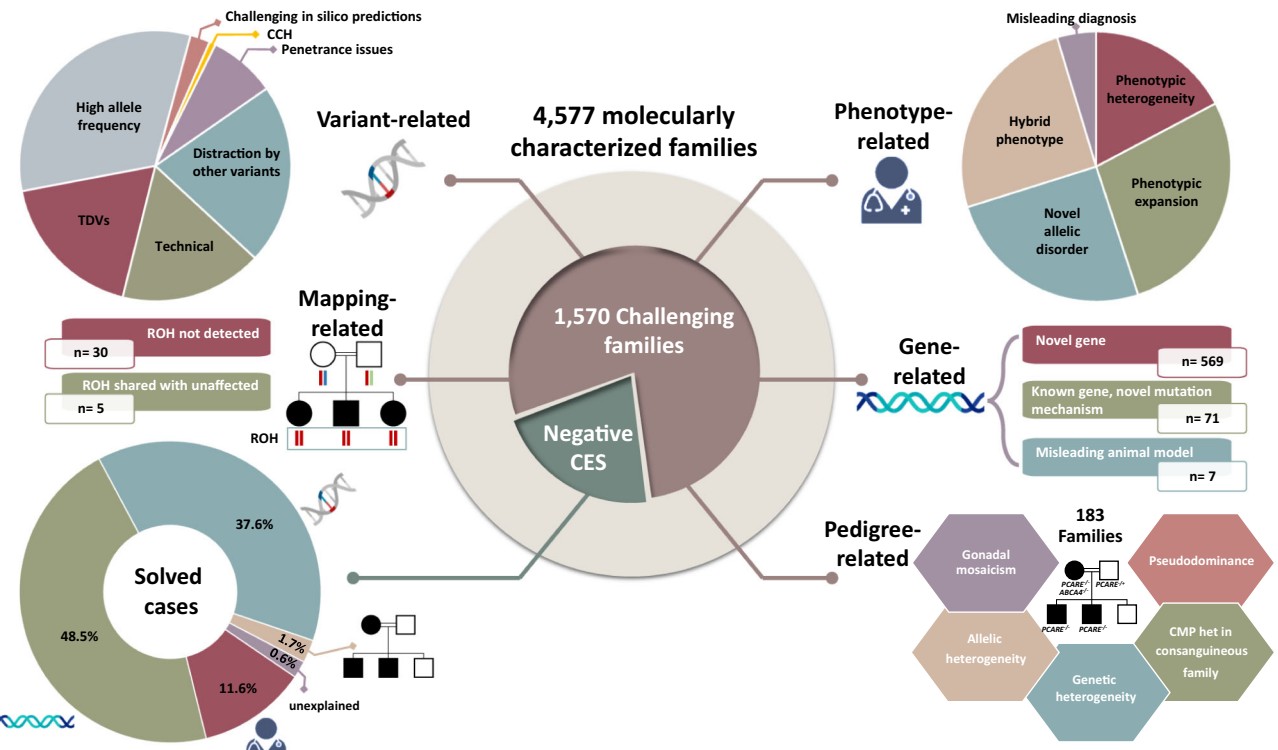

**Fig. 1 | Overview of the challenges encountered based on a cohort of 4577 molecularly characterized families.** In a cohort of 4577 molecularly characterized families, we encountered 5 main scenarios we found to be challenging in 1570 families. First, there are phenotype-related challenges that can be further sub-categorized into phenotypic heterogeneity, phenotypic expansion, novel allelic disorders, blended phenotype, and misleading diagnoses. Gene-related challenges comprise the discovery of novel disease genes, novel mutation mechanisms, and cases where the animal model did not corroborate the phenotype observed in human patients. Pedigree-related challenges comprise gonadal mosaicism, allelic

and genetic heterogeneity, and pseudo-dominance. Variant-related challenges comprise interpretation and technical level challenges. Pitfalls of autozygosity include the lack of detectable ROH at the disease locus and apparent sharing of the candidate ROH with an unaffected member of the family. We then dissect the prevalence of these challenges in 314 families referred to us with negative clinical exome or genome sequencing. We observed that most of the challenges encountered are either because the causal gene was novel at the time of analysis or because of non-technical variant related challenges. Logos for Gene-related, phenotype-related, and variant-related challenges are created using BioRender.com.

to have severe ovarian insufficiency with normal karyotype and *FMR1* repeat number. The novel homozygous variant in *TAL-DO1*:NM_006755.2:c.486_500dup;p.(His163_Thr167dup) in her exome was initially dismissed as irrelevant due to the lack of clinical features of transaldolase deficiency[10]. However, its association with elevated urinary excretion of erythritol, arabitol, and ribitol in the index, which is consistent with transaldolase deficiency, and its full segregation with the phenotype in the family prompted us to upgrade it to likely pathogenic and to propose a *TALDO1*-related ovarian insufficiency as a novel allelic disorder. Families F6581 and F6582 presented with fetal akinesia, and each had a different homozygous LOF allele in *COL25A1*. This strongly supports fetal akinesia as an allelic disorder distinct from *COL25A1*-related congenital fibrosis of extraocular muscles[11] (Fig. 2e, f). Family F4367 was clinically diagnosed with *HADHB*-related non-syndromic peripheral neuropathy, which is a distinct allelic disorder from the OMIM-listed *HADHB*-related trifunctional protein deficiency. Family F4607 is a simplex Bardet-Biedl syndrome case likely caused by a homozygous variant in *SCLT1*, a gene originally described in connection to oral-facial-digital syndrome[12]. We also highlight an apparently novel recessive allelic disorder caused by homozygous LOF in *VPS50* and comprises severe congenital hydrocephalus in family F9792 (no biallelic LOF variants have been reported before, which may explain the severe nature of this phenotype). Although *SCYL1*-related CALFAN syndrome (cholestasis, acute liver failure, and neurodegeneration) has been published[13], it is worth highlighting family F7600 with

this disease because this distinct allelic disorder remains unlisted in OMIM (only spinocerebellar ataxia is listed under *SCYL1*). Another interesting example is family F8309 with global developmental delay, epilepsy, and microcephaly. The patient was shown to have a heterozygous deletion chr1:238817161-249224684 removing *AKT3*. All previously reported variants in *AKT3* causing Megalencephaly-polymicrogyria-polydactyly-hydrocephalus syndrome 2 are missense gain of function variants so we propose the first loss of function variant causing microcephaly instead of megalencephaly as part of a new *AKT3*-related neurodevelopmental disorder.

iv. Blended phenotype: phenotypes caused by the presence of two or more Mendelian diseases in the same individual were observed in 87 families (5.5%, Supplementary Data 5, excluding ACMG secondary findings). Blended phenotypes are particularly challenging when multiple genes converge on the same phenotype. Examples include lissencephaly due to *LAMA2* and *CTSD* variants in family F270, inherited retinal degeneration due to *ABCA4* and *PCARE* in family F1895, anterior segment dysgenesis due to *PXDN* and *CYP1B1* variants in family F656, developmental and epileptic encephalopathy due to *SZT2* and *UGDH* variants in family F8406, and polycystic kidneys due to *HNF1B* and *PKD1* variants in family F6917 (Fig. 2g). It is worth highlighting that the blended phenotype need not be due to independently inherited variants in multiple genes. For example, the phenotype of hyperinsulinism and inherited retinal degeneration in families F4296, F6457, and F8752 was found to be caused by a founder deletion involving

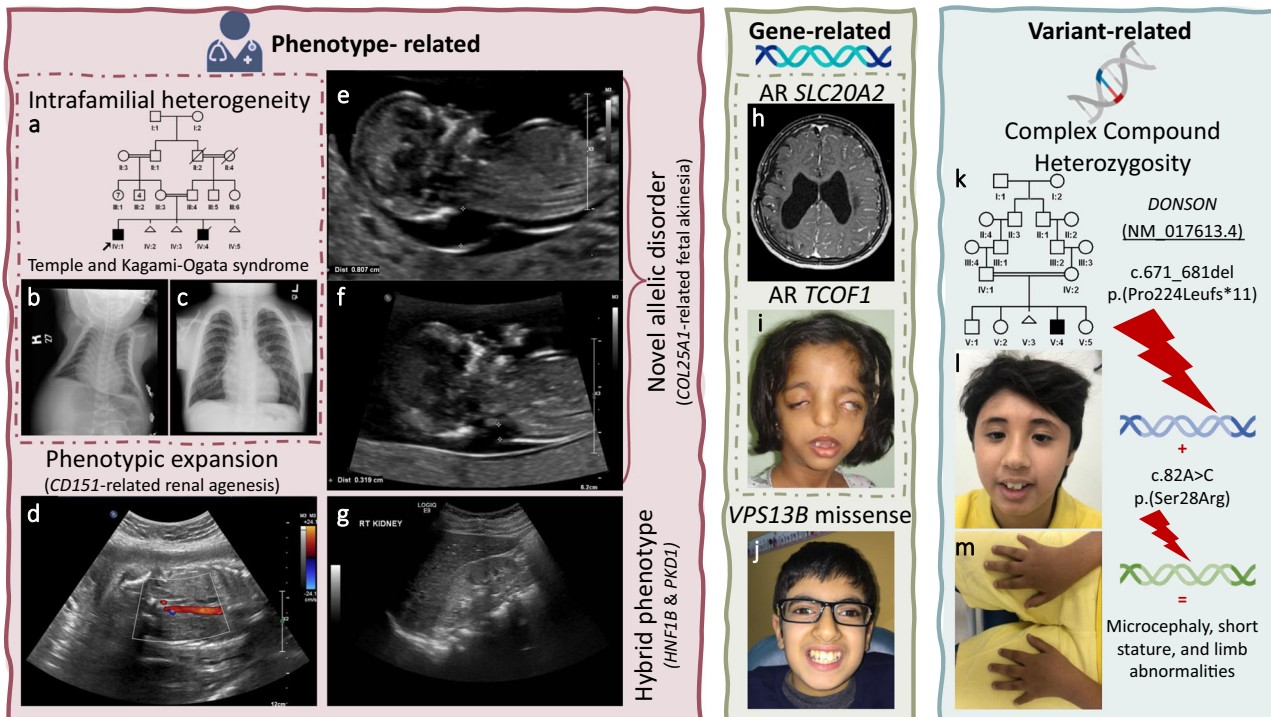

**Fig. 2 | Clinical images of select challenges. a–g** Phenotype-related challenges: **a** Pedigree of family F2640 with intrafamilial phenotypic heterogeneity molecularly diagnosed with Temple and Kagami-Ogata syndrome. **b** Chest X-ray of deceased individual IV:4 displaying "coat hanger" appearance. **c** Chest X-ray of individual IV:1 with the same pathogenic deletion on Chr 14 showing normal chest shape. **d** Phenotypic expansion: ultrasound imaging of fetus from family F7829 showing bilateral renal agenesis caused by a homozygous variant in *CD151*. **e**, **f** Novel allelic disorder: ultrasound imaging of fetuses from families F6581 and F6582, respectively, highlighting edema and *COL25A1*-related fetal akinesia. **g** Blended phenotype: ultrasound imaging of the right kidney of a child from family F6917 showing polycystic kidneys caused by two heterozygous variants in *HNF1B* and *PKD1*.

**h–j** Gene-related challenges: **h** MRI images of a brain of an affected individual from family F4159 molecularly diagnosed with biallelic variants in *SLC20A2* showing calcification. **i** Clinical images of an affected member of family F732 with a homozygous variant in *TCOF1*. **j** Clinical image of an affected child from family F3151 showing typical Cohen syndrome facies with the typical "grimace" upon smiling, molecularly diagnosed with a missense (rather than LOF) variant in *VPS13B*. **k–m** Variant-related challenge: **k** Pedigree of family F6211 with one child affected with Microcephaly, short stature, and limb abnormalities (**l**, **m**) caused by a compound heterozygous variant in *DONSON*. Logos for Gene-related, phenotype-related, and variant-related challenges are created using BioRender.com.

*ABCC8* and *USH1C*. Remarkably, one patient (F8752) presented with this phenotype and tested negative for this deletion. Instead, he was found to have independent homozygous pathogenic variants in *ABCC8* and *LCA5*. Perhaps the most striking examples are families F900 and F8348, each molecularly and clinically diagnosed with three different diseases in the same individual. Family F900 was referred to us with albinism and was found to be homozygous for Hermansky-Pudlak syndrome-related variant *HPS4*:ENST00000398145.2:c.502-1 G > A and glucose/galactose malabsorption-related variant *SLC5A1*:NM_000343.4:c.765 C > G; p.(Cys255Trp) and hemizygous for the founder variant for hemolytic anemia *G6PD*:ENST00000393562.2:c.233 T > C; p.(Ile78Thr). Family F8348 was referred with a complex phenotype (sick baby in NICU with polycystic kidney disease and a very thick scleroderma-like skin in addition to multi-organ failure). Molecular analysis revealed homozygous variants in *RAG1*:NM_000448.3:c.554delG;p.(Lys186Serfs*15) and *TXNDC15*: NM_024715.4:c.703 C > T;p.(Arg235Trp) fully explaining the individual's immunodeficiency and ciliopathy phenotypes, respectively. He was also found to be heterozygous for epidermolysis bullosa-related variant *KRT14*:NM_000526.5:c.915 G > A;p. (Trp305*) which may be related to his skin phenotype.

v. Erroneous clinical labels: we encountered 15 families (~1%, Supplementary Data 6) in which the wrong diagnostic label caused a major delay in the molecular diagnosis. For example, in family F8078, a genetic diagnosis was initially dismissed because the brain MRI findings were typical of hypoxic-ischemic

encephalopathy. However, a microduplication spanning *CACNA1B* gene and a part of *EHMT1* gene was subsequently identified.

vi. Non-Mendelian phenotypes: These include neurodevelopmental disorders that turned out to be caused by environmental factors unknown at the time of recruitment and familial clustering of complex phenotypes that masquerade as Mendelian phenocopies. One notable example is the child with arthrogryposis multiplex (F6) and maternal history of multiple miscarriages. It was later found that the mother suffers from an antiphospholipid syndrome which likely explains the phenotype. This category accounted for 28 families (1.8%).

**2- Gene-related:**

i. Novel gene-disease assertions: When the gene-disease assertion was novel at the time of analysis, the molecular diagnosis was greatly delayed. Apart from 83 families affected with 52 novel allelic disorders reported above, we highlight 132 gene-disease assertions that are not yet listed in OMIM (Supplementary Data 7). Similarly, variants in genes with only questionable gene-disease assertion at the time of analysis were challenging to interpret and this scenario was encountered in 155 families (9.8%, Supplementary Data 7). The latter table can be considered a resource to support previously proposed gene-disease assertion and it includes 16 unpublished families.

ii. Incompatible phenotype in the animal model: when considering novel gene-disease assertions, the phenotype of the animal model

**Table 1 | Families with novel allelic disorders described in this manuscript for the first time**

| Pedigree | ID number | Disease Name | Phenotype-specific OMIM ID | Gene | Variant | Zygosity |
|---|---|---|---|---|---|---|
| F141 | 08DG-00413 | OTX2-related retinal degeneration with female infertility | Not listed | OTX2 | NM_001270525.2:r.98_273del;p.(Pro34Metfs*3) | Homozygous |
| F2179 | 11DG1507 | ZFHX4-related cone rod dystrophy | Not listed | ZFHX4 | NM_024721.5:c.5020 C > T;p.(Gln1674*) | Heterozygous |
| F2593 | 12DG0172 | IFT140-related LCA | Not listed | IFT140 | NM_014714.4:c.1990G>A;p.(Glu664Lys) | Homozygous |
| F2924 | 12DG1193 | NOTCH2-related retinitis pigmentosa | Not listed | NOTCH2 | NM_024408.4:c.1619C>T;p.(Pro540Leu) | Homozygous |
| F4143 | 14DG0621 | UNC80-related arthrogryposis | Not listed | UNC80 | NM_032504.2:c.7948 G > A;p.(Gly2650Arg) | Homozygous |
| F4367 | 14DG1509 | HADHB-related non-syndromic neuropathy | Not listed | HADHB | NM_000183.2:c.712 C > T;p.(Arg238Trp) | Homozygous |
| F454 | 09DG00739 | NOTCH2-related retinitis pigmentosa | Not listed | NOTCH2 | NM_024408.4:c.1619C>T;p.(Pro540Leu) | Homozygous |
| F4607 | 14DG2149 | SCLT1-related BBS | Not listed | SCLT1 | NM_144643.4:c.778_780del;p.(Glu260del) | Homozygous |
| F5409 | 15DG2563 | FZD6-related nonimmune hydrops fetalis | Not listed | FZD6 | ENST00000358755.4:c.869 A > G;p.(Tyr290Cys) | Homozygous |
| F6581 | 19DG0549 | COL25A1-related fetal akinesia | Not listed | COL25A1 | NM_198721.4:c.1598delC;p.(Pro533Hisfs*77) | Homozygous |
| F6582 | 19DG0552 | COL25A1-related fetal akinesia | Not listed | COL25A1 | Large deletion involving COL25A1 and ZCCHC23 | Homozygous |
| F6666 | 19DG0767 | ABL1-related mirror image of CHDSKM | Not listed | ABL1 | NM_005157.6:c.1966_2011dupCCAGCCAAGTCCCCAAAGCCCAGCAATGGGGTCTGGGGTCCCCAATG;p.(Gly671Alafs*93) | Homozygous |
| F7600 | 19DG2397 | CALIFAN syndrome | Not listed | SCYL1 | NM_020680.4:c.1386+1 G > T | Homozygous |
| F8309 | PSMMC-0377 | AKT3-related NDD with microcephaly | Not listed | AKT3 | Large deletion chr1:238817161-249224684 involving ZBTB18, HNRNPU, and AKT3 | Heterozygous |
| F8485 | PSMMC-0385 | KCNMA1-related NDD | Not listed | KCNMA1 | Chr10:79364485-79611505 duplication spanning KCNMA1 | Heterozygous |
| F9597 | 22DG1526 | RHOBTB2-related neuro-developmental disorder | Not listed | RHOBTB2 | NM_015178.3:c.394 C > T;p.(Arg132*) | Homozygous |
| F9792 | Fetus of 23DG1304 | VPS50-related hydrocephalus | Not listed | VPS50 | NM_017667.4:c.1705C>T;p.(Arg569*) | Homozygous |
| F4216 | 14DG0915 | COL2A1-related arthrogryposis | Not listed | COL2A1 | NM_001844.5:c.985 C > T;p.(Pro329Ser) | Homozygous |
| F8629 | 21DG0260 | TALDO1-related ovarian insufficiency | Not listed | TALDO1 | NM_006755.2:c.486_500dup;p.(His163_Thr167dup) | Homozygous |

is an important consideration. Nonetheless, there are instances where the reported phenotype in the animal model is sufficiently incompatible to delay the establishment of the gene-disease assertion. For example, *MPDZ*-related congenital hydrocephalus seen in families F2268, F2699, and F5606 was initially dismissed because a chick model lacked hydrocephalus[14]. It was only after the original publication of *MPDZ*-related hydrocephalus[15] that a compatible mouse model recapitulating the human phenotype was published[16]. Additional examples include the animal models of *MICU2* and *AGBL5* deficiency identified[17,18], respectively, in family F5563 with severe neurodevelopmental disorder and family F2707 with non-syndromic retinal dystrophy. Supplementary Data 8 details this category that potentially affected 7 families (0.45%).

iii. Known gene, novel mutation mechanism: the likely causal variant was dismissed in 71 families (4.5%) because of the perceived incompatibility of the identified homozygous variant with the established dominant inheritance pattern of the implicated gene. Supplementary Data 9 lists these cases, including 23 unpublished (Table 2). As compared to the typical dominant phenotype, the recessive phenotype ranged from similar e.g. *SLC20A2*-related Basal ganglia calcification in family F4159 (Fig. 2h), *TCOF1*-related Treacher-Collin syndrome in family F732 (Fig. 2i) and *MAPRE2*-related circumferential skin creases in family F531, to more severe e.g. *ADSS1*-related embryonic lethality in family F8399, to distinct allelic disorders. One remarkable example of the latter is family F6666 in which two siblings presented with a dysmorphic syndrome and associated global developmental delay (Fig. 3a). The biallelic loss of function variant in *ABL1* these two siblings share is very different from the gain of function de novo variants identified in *ABL1*-related congenital heart defects and skeletal malformations syndrome CHDSKM. The dysmorphology profile can be viewed as the opposite of CHDSKM, which has been likened to Van de Ende Gupta syndrome[19] (Fig. 3b–h). A similarly remarkable example is family F141 where an *OTX2*-related inherited retinal degeneration with female infertility caused by a homozygous cryptic splicing variant is a stark contrast to the dominant *OTX2*-related anophthalmia phenotype (Supplementary Figure 1A and B). A third example is F454 and F2924, two families in which a non-syndromic retinitis pigmentosa phenotype fully segregated with a founder recessive *NOTCH2* variant even though this gene is only linked to very distinct autosomal dominant conditions in OMIM. Family F3151 with dysmorphia was molecularly diagnosed with a missense (rather than LOF) variant in *VPS13B* (Fig. 2j). Another example is two families F7810 and F9597, where we identified a homozygous founder LOF variant in *RHOBTB2* leading to a *RHOBTB2*-related neurodevelopmental disorder characterized by global developmental delay, mild facial dysmorphism, normal brain MRI and no epilepsy in stark contrast to the *RHOBTB2*-related developmental and epileptic encephalopathy caused by de novo gain of function variants.

### 3- Variant-related:

i. Interpretation challenges:

a. Tentative transcript-deleterious variants (TDV): With the exception of canonical ±1/2 splicing donors and acceptors, all other tentative TDV were challenging to interpret even though they were mostly compatible with exome capture. These accounted for 177 families (11.3%, Supplementary Data 10 includes 85 families described in Maddirevula et al., and 92 described here for the first time). Notable examples include families F2204, F7801, and F8083 with severe lactic acidosis that defied analysis until the synonymous variant *BCS1L*:NM_001079866.2:c.441 C > T;p.(=) was found to be TDV. This variant was particularly challenging because it is synonymous and does not appear to be splicing in nature (deep exonic). However, RT-PCR experiments revealed aberrant splicing resulting in frameshift and early truncation of the protein (*BCS1L*:NM_001079866.2:c.441 C > T;r.436_460delGTTTTCTTCAA-CATCCTGGAGGAAG;p.(Val146Leufs*4)).

b. Allele frequency above cut-off: a default AF of <0.001 is often used as a threshold for rare recessive diseases[20]. However, variants with AF above cut-off accounted for the molecular diagnosis in 255 families (16.2%, Supplementary Data 11 and Supplementary Fig. 2). While some of these high AF variants posed no challenge because they cause diseases known to be very common e.g. sickle cell anemia, congenital glaucoma, and Gilbert syndrome, many were challenging to assign a pathogenic role, and these can be grouped into the following:

1. The variant is a founder variant that causes a disease with a previously unrecognized relatively high incidence, e.g. *MPL*-related Thrombocythemia 2, *WASHC5*-related Ritscher-Schinzel syndrome 1, *PTRHD1*-related non-syndromic intellectual disability, *NT5DC1*-related intellectual disability, and *RECQL4*-related RAPADILINO syndrome (AR). These variants are being highlighted as major founders in the population for the first time with local AF of 0.018073, 0.001127, 0.002435, 0.001127, and 0.001623, respectively.

2. The variant is only pathogenic when compound heterozygous with a more damaging variant: this is the case for *RBM8A*:NM_005105.5:c.*6 C > G (AF of 0.014411), *SBDS*:NM_016038.4:c.258+2 T > C (AF of 0.001082), *DHCR7*:NM_001360.3:c.1 A > G;p.? (AF of 0.003652), *POLR3A*:NM_007055.4:c.1909+22 G > A (AF of 0.005418) and *EYS*:NM_001142800.2:c.2137+1 G > A (AF of 0.007348) associated with thrombocytopenia-absent radius syndrome, Shwachman-Diamond syndrome 1, Smith-Lemli-Opitz syndrome, Hypomyelinating leukodystrophy, and Retinitis pigmentosa 25, respectively (see complex compound heterozygous inheritance below).

3. The variant leads to a highly variable phenotype: an example worth highlighting is *ALDOB*:NM_000035.4:c.448 G > C;p.(Ala150Pro) with local AF of 0.002705, which was identified in a 63-year-old male who presented to the clinic for a familial cancer phenotype. ES analysis revealed that the index individual and, subsequently by Sanger analysis, several other family members have a homozygous variant in *ALDOB* known to cause hereditary Fructose intolerance (Supplementary Fig. 1c). This finding is highly unusual because the index patient and other homozygous adults did not have any liver complications and only volunteered sweet aversion on further questioning.

4. The variant causes a common but embryonic lethal disease, e.g. *CHRNA1*:NM_000079.4:c.254 T > C;p.(Leu85Pro) with AF of 0.001262.

5. The variant causes a common but cryptic phenotype: one dramatic example is *TGM5*-related acral peeling syndrome caused by NM_201631.4:c.1335 G > C;p.(Lys445Asn). Interestingly, clinical WGS sequencing was reported negative in F9185 with this variant because it was classified internally as benign based on the high local AF of 0.010242 and its presence in "homozygous individuals with no reported skin phenotype". However, this is a previously reported founder variant[21] and, as in the previous report, our patient has the classical clefting between stratum corneum and stratum granulosom (Supplementary Fig. 3). The highly variable nature of this condition, especially in the local hot environment that induces occasional acral blistering during excessive sweating even among normal individuals likely accounts for the cryptic nature of this condition. Another example *C8B*:NM_000066.4:c.1282 C > T;p.(Arg428*), which causes an immunodeficiency that only manifests under certain

**Table 2 | Cases with known gene, novel mutation mechanisms described in this manuscript for the first time**

| Pedigree | ID number | Gene | Disease Name | Phenotype-specific OMIM ID | Variant | Zygosity | Known gene, novel mutation mechanism |
|---|---|---|---|---|---|---|---|
| F141 | 08DG-00413 | OTX2 | OTX2-related inherited retinal degeneration with female infertility | Not listed | NM_001270525.2:r.98_273del;p.(Pro34Metfs*3) | Homozygous | Biallelic rather than monoallelic |
| F2924 | 12DG1193 | NOTCH2 | NOTCH2-related retinitis pigmentosa | Not listed | NM_024408.4:c.1619C>T;p.(Pro540Leu) | Homozygous | Biallelic rather than monoallelic |
| F3108 | 12DG1912 | FOXE3 | Anterior segment dysgenesis 2, multiple subtypes | 610256 | NM_012186.3:c.720 C > A;p.(Cys240*) | Homozygous | Biallelic rather than monoallelic |
| F3151 | 12DG2122 | VPS13B | Cohen syndrome | 216550 | NM_152564.5:c.5783 C > T;p.(Pro1928Leu) | Homozygous | Missense rather than LOF |
| F4370 | 14DG1521 | RIMS1 | Cone-rod dystrophy 7 | 603649 | ENST00000521978.1:c.3143 T > G;p.(Leu1048*) | Homozygous | Biallelic rather than monoallelic |
| F4422 | 14DG1734 | AKT3 | Megalencephaly-polymicrogyria-polydactyly-hydrocephalus syndrome 2 | 615937 | NM_005465.7:c.1393 C > T;p.(Arg465Trp) | Heterozygous (De novo) | Constitutional rather than mosaic |
| F454 | 09DG00739 | NOTCH2 | NOTCH2-related retinitis pigmentosa | Not listed | NM_024408.4:c.1619C>T;p.(Pro540Leu) | Homozygous | Biallelic rather than monoallelic |
| F4735 | 15DG0159 | COL9A3 | Stickler syndrome, type VI | 620022 | NM_001853.4:c.754 C > T;p.(Arg252*) | Homozygous | Biallelic rather than monoallelic |
| F531 | 09DG00980 | MAPRE2 | Symmetric circumferential skin creases, congenital, 2 | 616734 | NM_001384732.1:c.815Q_815IdelGA;p.(Gly2717Alafs*40) | Homozygous | Biallelic rather than monoallelic |
| F5683 | 16DG1189 | MYH9 | Macrothrombocytopenia and granulocyte inclusions with or without nephritis or sensorineural hearing loss | 155100 | NM_002473.6:c.2206 G > A;p.(Gly736Arg) | Homozygous | Biallelic rather than monoallelic |
| F6666 | 19DG0767 | ABL1 | ABL1-related mirror image of CHDSKM | Not listed | NM_005157.6:c.1966_2011dupCCAGCCAAGTCCCCAAAGCCCAGCAATGGGGCTGGGGTCCCCAATG;p.(Gly671Alafs*93) | Homozygous | Biallelic LOF rather than de novo GOF |
| F8175 | PSMMC-0351 | PHOX2B | Central hypoventilation syndrome, congenital, 1, with or without Hirschsprung disease | 209880 | NM_003924.3:c.590del;p.(Gly197Alafs*112) | Heterozygous | De novo LOF rather than repeat expansion |
| F8309 | PSMMC-0377 | AKT3 | AKT3-related NDD with microcephaly | Not listed | Large deletion chr1:238817161-249224684 involving ZBTB18, HNRNPU, and AKT3 | Heterozygous | LOF rather than GOF |
| F8399 | 20DG1091 | ADSS1 | Myopathy, distal, 5 | 617030 | NM_152328.5:c.1073+1 -> T | Homozygous | LOF rather than missense |
| F8485 | PSMMC-0385 | KCNMA1 | KCNMA1-related NDD | Not listed | 10q22.3(79364485-79611505)x3 | Heterozygous | Triplo-sensitivity |
| F9597 | 22DG1526 | RHOBTB2 | RHOBTB2-related neuro-developmental disorder | Not listed | NM_015178.3:c.394 C > T;p.(Arg132*) | Homozygous | Biallelic rather than monoallelic |
| F9792 | Fetus of 23DG1304 | VPS50 | VPS50-related hydrocephalus | Not listed | NM_017667.4:c.1705C>T;p.(Arg569*) | Homozygous | LOF rather than missense |

**Table 2 (continued) | Cases with known gene, novel mutation mechanisms described in this manuscript for the first time**

| Pedigree | ID number | Gene | Disease Name | Phenotype-specific OMIM ID | Variant | Zygosity | Known gene, novel mutation mechanism |
|---|---|---|---|---|---|---|---|
| F9145 | PSMMC0442 | MPL | Thrombocythemia 2 | 601977 | NM_005373.3:c.317 C > T;p.(Pro106Leu) | Homozygous | Biallelic rather than monoallelic |
| F6329 | 19DG2474 | SCN3A | Epilepsy, familial focal, with variable foci 4 | 617935 | NM_006922.4:c.3003 G > C;p.(Met1001Ile) | Homozygous | Biallelic rather than monoallelic |
| F7988 | 20DG0276 | ACVR2B | Heterotaxy, visceral, 4, autosomal | 613751 | NM_001106.4:c.92 A > G;p.(Tyr31Cys) | Homozygous | Biallelic rather than monoallelic |
| F9176 | PSMMC9176 | CRYBA4 | Cataract 23 | 610425 | NM_001886.3:c.206 T > C;p.(Leu69Pro) | Homozygous | Biallelic rather than monoallelic |
| F4216 | 14DG0915 | COL2A1 | COL2A1-related arthryposis | Not listed | NM_001844.5:c.985 C > T;p.(Pro329Ser) | Homozygous | Biallelic rather than monoallelic |
| F5095 | 15DG1215 | CREBBP | Rubinstein-Taybi syndrome 1 | 180849 | Multi-exon duplication (4–11) | Homozygous | Biallelic rather than monoallelic |

microbial exposure (AF 0.002615). The male infertility-related variants identified in *SPATA3, TTC21A, TERB1, HYDIN,* and *CCDC155* can be considered additional examples.

6. No clear reason for the discrepancy between the high AF and the observed disease phenotype, e.g. *DYNC2H1*:NM_001377.3:c.1151 C > T;p.(Ala384Val) which is associated with Short-rib thoracic dysplasia 3 with or without polydactyly with local AF of 0.001397 and *DALRD3*:NM_001009996.3:c.1251 C > A;p.(Tyr417*) associated with Developmental and epileptic encephalopathy 86 (AF 0.001082).

a. Challenging in silico prediction: We have encountered 70 families (4.5%, Supplementary Data 12) where the causal variant had poor in silico prediction scores. The likely causal variant may even be fixed in other organisms e.g. *EZH2*:NM_004456.4:c.2233 G > A;p.(Glu745Lys) variant in family F1294 with Weaver syndrome was difficult to interpret because it corresponds to the normal allele in marmosets. Poor in silico predictions at the protein level of 8 missense variants belied their deleterious nature at the RNA level (see tentative TDVs). Non-coding genes represented a distinct class of in silico prediction challenges e.g. *RNU4ATAC*-related primordial dwarfism.

b. Complex compound inheritance (Table 3): This phenomenon refers to recessive variants that only cause disease when inherited in trans with a different variant (see Allele frequency above cut-off category above). Family F1865 with Stargardt disease is a good example where the *ABCA4* variant NM_000350.3:c.5882 G > A;p.(Gly1961 Glu) was initially dismissed because of the high frequency in gnomAD in addition to the presence of unaffected homozygous individuals. Of note, functional studies have demonstrated a deleterious nature of this variant[22,23]. We suggest this is a mild variant only pathogenic in trans with a more severe variant as in this family where it was inherited in trans with a strong loss of function allele NM_000350.3:c.1937+1G > A. Another example is *POLR3A*: NM_007055.4:c.1909+22 G > A that has been shown to cause aberrant splicing[24]. However, this variant is present five times as homozygous in our database in individuals who lack the respective OMIM phenotype. Therefore, and in view of the compound heterozygous nature of previously published cases, we suggest this variant is pathogenic only when in trans with a more severe variant. Another example is family F6211 with an affected child with Microcephaly, short stature, and limb abnormalities syndrome caused by compound heterozygous variants (NM_017613.4:c.671_681del;p.(Pro224Leufs*11)) and c.82 A > C;p.(Ser28Arg)) in *DONSON* (Fig. 2k–m). The missense variant was previously suggested to have no effect on protein stability and sub-cellular localization[25]; however, we hypothesize that if the variant is present in trans with a strong allele, it can be disease-causing. It should be noted that we cannot completely rule out the possibility that homozygotes of the above highlighted alleles may rarely manifest clinically.

c. Multivariant alleles: When more than one pathogenic variant is detected in the same allele, it may be difficult to conclude if all are contributing to the phenotype. For example, we have identified the two variants *CNGA3*:NM_001298.3:c.101+1 G > A and NM_001298.3:c.661 C > T;p.(Arg221*) in multiple families with achromatopsia. However, we later identified another affected family that is only homozygous for the c.101+1 G > A on the same haplotype background indicating that this is the ancestral disease haplotype and is sufficient to cause the disease.

d. Incomplete penetrance: In 65 families (4.1%, Supplementary Data 13), the causal variant was difficult to interpret due to a perceived lack of segregation (including the presence of homozygous state in public or local databases), which posed a

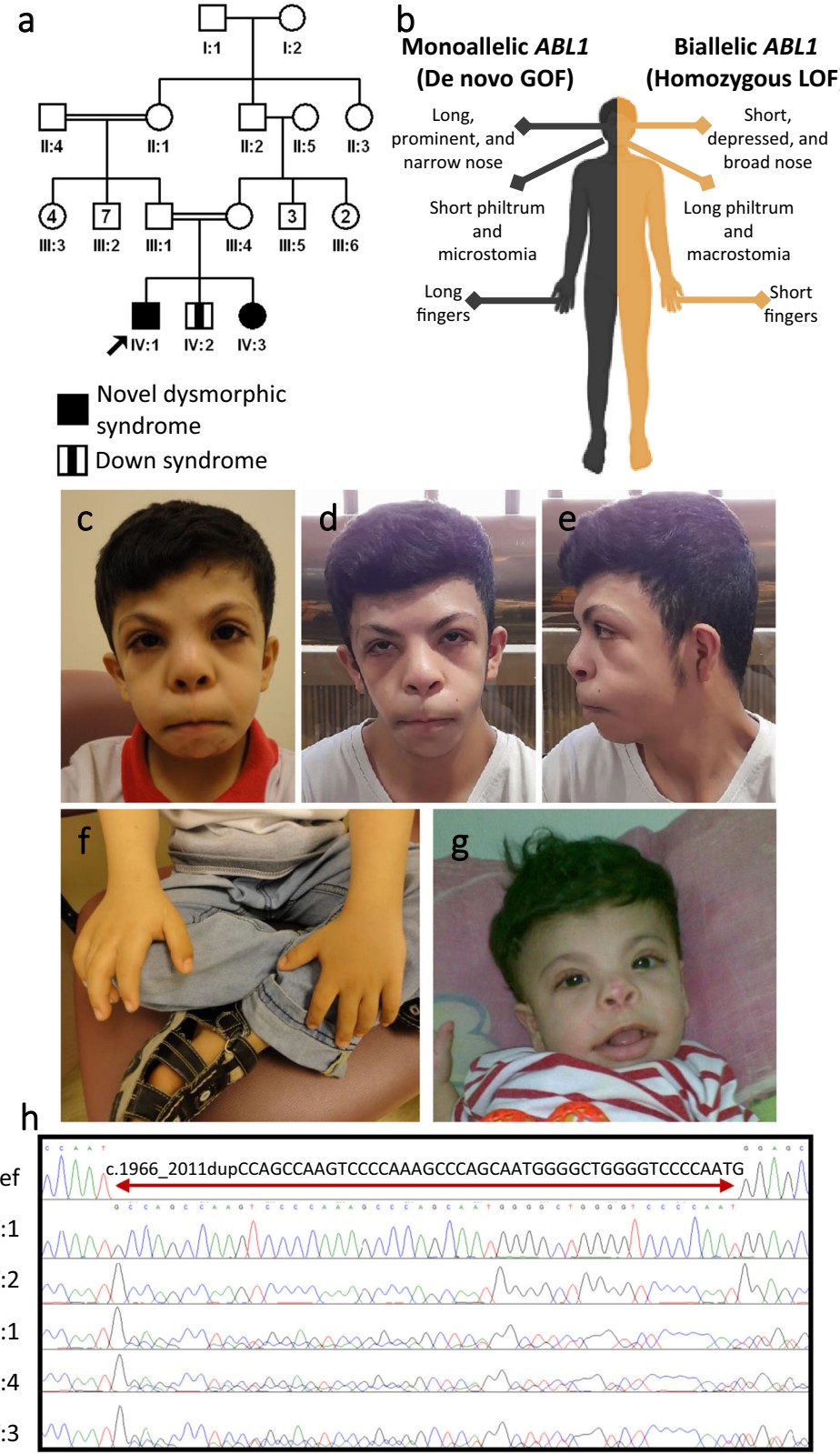

**Fig. 3 | A novel allelic disorder caused by biallelic LOF in *ABL1*. a** Pedigree of family F6666 with two similarly affected children and one child affected with Down syndrome. **b** Schematic representation contrasting the phenotypic differences between monoallelic gain of function and biallelic loss of function variants in *ABL1*. **c**–**e** Clinical photographs showcasing facial dysmorphia in affected individual IV:1. **f** Photograph showing short fingers in individual IV:1. **g** Clinical photograph of the similarly affected sister IV:3. **h** Sanger sequencing showing the homozygous 46 bp duplication in *ABL1* in the two affected siblings, which is heterozygous in the parents and brother affected with Down syndrome. Panel (**b**) was created with BioRender.com.

**Table 3 | Variants displaying complex compound heterozygosity**

| Gene | Variant | Justification for CCH | Disease Name | Phenotype-specific OMIM ID |
|---|---|---|---|---|
| POLR3A | NM_007055.4:c.1909+22 G > A | Homozygous in individuals who lack the phenotype | Leukodystrophy, hypomyelinating, 7, with or without oligodontia and/or hypogonadotropic hypogonadism | 607694 |
| ABCA4 | NM_000350.3:c.5882 G > A;p.(Gly1961Glu) | Homozygous in individuals who lack the phenotype | Stargardt disease 1 | 248200 |
| SBDS | NM_016038.4:c.258+2 T > C | Homozygous in individuals who lack the phenotype | Shwachman-Diamond syndrome 1 | 260400 |
| RBM8A | NM_005105.5:c.*6 C > G | Previously published CCH | Thrombocytopenia-absent radius syndrome | 274000 |
| EYS | NM_001142800.1:c.2137+1 G > A | Homozygous in individuals who lack the phenotype | Retinitis pigmentosa 25 | 602772 |
| DHCR7 | NM_001360.3:c.1 A > G;p.? | Homozygous in individuals who lack the phenotype | Smith-Lemli-Opitz syndrome | 270400 |
| DONSON | NM_017613.4:c.82 A > C;p.(Ser28Arg) | Homozygous in individuals who lack the phenotype | Microcephaly, short stature, and limb abnormalities | 617604 |
| CLCN7 | NM_001287.6:c.1208 G > A;p.(Arg403Gln) | Homozygous in individuals who lack the phenotype | Osteopetrosis, autosomal recessive 4 | 611490 |

challenge until it was later realized that this was a penetrance issue. Age-related penetrance was most common and was observed in 12 families, typically related to progressive retinal degeneration. Pathogen-specific penetrance was observed in families F8448 and F5452 with *C8B* and *MSN* variants, respectively. We also observed sex-related penetrance e.g. in the case of F8606 with testicular regression syndrome, the identified variant in *DHX37* was also initially dismissed due to its presence in unaffected members before realizing this was a sex-limited phenotype. However, no explanation could be found for the segregation results in some cases. For example, the index patient in family F9182 was found to have the CNV (1:145390101_145786290 heterozygous deletion) inherited from a healthy father even though it was reported pathogenic in 7 individuals with overlapping neurodevelopmental disorders in DECIPHER[26]. Reduced penetrance is well established for CNVs[27] but unusual in autosomal recessive SNVs. Thus, family F1028 was particularly surprising given the apparent non-penetrance of a homozygous LOF variant in a typically fully penetrant morbid gene. Specifically, a homozygous truncating variant in *DENND5A*:NM_015213.4:c.3387+3 G > T;r.3305_3387del;-p.(Lys1102Thrfs*27) was initially ignored in three affected siblings with the typical *DENND5A*-related developmental and epileptic encephalopathy phenotype because it was also homozygous in their unaffected sibling (Supplementary Fig. 3). The reason for incomplete penetrance in this family remains obscure and we suspect this may be a rare form of resilience (see Discussion). On the other hand, apparent non-penetrance can be explained by inadequate phenotyping. This can be seen when the patient is difficult to phenotype due to a complex presentation. For example, the classical Congenital symmetric circumferential skin creases syndrome was overlooked in a child (family F531) with a pathogenic variant in *MAPRE2* because he also suffered from *CPLANE1*-related Joubert syndrome, but it could readily be observed in the sister who is only homozygote for the *MAPRE2* variant. In some cases, the phenotype is observed but not perceived as an extension of the phenotype under study in the index. An example is F5409, where NIHF (non-immune hydrops fetalis) was investigated. One sister was initially recruited as unaffected because she only had club nails which were later found to be an extension of the *FZD6*-related NIHF phenotype (Supplementary Fig. 3). We suspect inadequate phenotyping may also play a role in explaining the presence of homozygotes for some pathogenic variants in gnomAD e.g. there is a homozygote in gnomAD for the pathogenic stopgain variant we identified in *KATNIP* in family F6141 with classical Joubert syndrome.

e. Distraction by other variants: In 240 families (15.3%, Supplementary Data 14), the search for the causal variant was derailed by another variant. These "red herring" variants can be grouped as follows:

i. Presumptive loss of function variants in known disease-related genes: These were obviously considered compelling candidates but were later found to be non-disease causing. We have encountered this phenomenon in 58 families (3.7%, Table 4 and Supplementary Data 14). The reasons for these variants not being disease-causing include:

1. LOF is not a disease mechanism: For example, the stopgain variant in *COL8A2*:NM_005202.4:c.1815C>A;p.(Tyr605*) was identified in three families that lacked corneal dystrophy. All *COL8A2*-related corneal dystrophy variants to date are amino acid substitutions and pLI is low (0.12).

2. The variant is not true LOF: For example, *EYS*: NM_001142800.1:c.2137+1 G > A was never shown to cause LOF on RT-PCR. Its high AF in gnomAD suggests it is not disease-causing, at

**Table 4 | Presumptive loss of function variants in known disease-related genes that are not disease-causing[†]**

| Gene | Variant | Zygosity | Disease name | Phenotype-specific OMIM ID | Reason to question pathogenicity | Proposed mechanism for non-pathogenicity |
|---|---|---|---|---|---|---|
| COL8A2 | NM_005202.4:c.1815C>A;p.(Tyr605*) | Homozygous | Corneal dystrophy, Fuchs endothelial, 1 | 136800 | Updated high MAF | Pathogenicity is due to GOF rather than LOF[59] |
| DENND5A | NM_015213.4:c.3387+3G > T;r.3305_3387del;p.(Lys1102Thrfs*27) | Homozygous | Developmental and epileptic encephalopathy 49 | 617281 | Homozygous in an unaffected sibling | ?Resilience |
| RPGR | NM_000328.3:c.1905+1G > A | Hemizygous | Cone-rod dystrophy, X-linked, 1 | 304020 | Homozygous in healthy individuals | The variant is a missense variant in the canonical transcript while +1 in all others |
| LIPN | NM_001102469.2:c.302delG;p.(Gly101Glufs*7) | Homozygous | Ichthyosis, congenital, autosomal recessive 8 | 613943 | Homozygous in an unaffected sibling | Gene-disease assertion in OMIM is based on one family |
| IQSEC1 | NM_001134382.3:c.2978del;p.(Pro993Hisfs*127) | Homozygous | Intellectual developmental disorder with short stature and behavioral abnormalities | 618687 | Homozygous in healthy individuals | The deleted bp is the last bp in the exon-intron boundary and the intron is only one base pair long |
| NPHP4 | NM_015102.5:c.2818-2 T > A | Homozygous | Nephronophthisis 4 | 606966 | Updated high MAF | Previous study has shown that the aberrant transcripts do not cause frameshift truncation[28] |
| GNAT1 | NM_144499.3:c.858 C > G;p.(Tyr286*) | Homozygous | Night blindness, congenital stationery, type 1 G | 616389 | Homozygous in an unaffected father | Truncates ~20% of the protein where no deleterious variants are reported |
| OFD1 | NM_003611.3:c.2600-1 G > C | Hemizygous | Orofaciodigital syndrome I | 311200 | Hemizygous in an unaffected father | The variant is not a LOF variant (does not cause a frameshift truncation) |
| EVC | NM_001306090.2:c.2731 C > T;p.(Arg911*) | Homozygous | Ellis-van Creveld syndrome | 225500 | Updated high MAF | Truncates ~9% of the protein |
| GUCA1A | NM_001384910.1:c.235delC;p.(Ala79Glyfs*23) | Homozygous | Cone dystrophy-3 | 602093 | Homozygous in healthy individuals | Pathogenicity is due to GOF rather than LOF[60] |
| MKS1 | Exon 1 loss at the RNA level | Homozygous | Bardet-Biedl syndrome 13 | 615990 | Alternative etiology was identified | Technical error |
| LETM1 | NM_012318.3:c.1678A>T;p.(Lys560*) | Homozygous | Neurodegeneration, childhood-onset, with multisystem involvement due to mitochondrial dysfunction | 620089 | Homozygous in an unaffected sibling | ?Resilience |
| LZTFL1 | NM_001276379.1:c.3 G > A;p.? | Homozygous | Bardet-Biedl syndrome 17 | 615994 | Updated high MAF | Startloss in some transcript and UTR variant in others. GTEX database reported that the shorter transcripts (where the variant is located in the UTR regions) are the ones abundantly expressed in most tissues especially the brain and reproductive organs |
| PDE11A | NM_016953.3:c.1633C>T;p.(Arg545*) | Homozygous | Pigmented nodular adrenocortical disease, primary, 2 | 610475 | Homozygous in an unaffected sibling | Pathogenicity is due to GOF rather than LOF[61] |
| RP1L1 | NM_178857.6:c.5959 C > T;p.(Gln1987*) | Homozygous | Retinitis pigmentosa 88 | 618826 | Updated high MAF | Truncates a small part of the C-terminus where no deleterious variants are reported |
| CDKL5 | NM_003159.2:c.2854 C > T;p.(Arg952*) | Heterozygous | Developmental and epileptic encephalopathy 2 | 300672 | Updated high MAF | Truncates 7.7% of the protein |

[†]Variants involved in complex compound heterozygosity are listed in Table 3.

least not in the homozygous state we encountered in 6 families. It is possible that this variant is involved in the complex compound inheritance phenomenon (see above). Another example is *RPGR*:NM_000328.3:c.1905+1 G > A, which we found hemizygous in three families with no evidence of retinal dystrophy. The reason for the non-pathogenicity of this variant is unclear but it is worth highlighting that its predicted impact on splicing was not experimentally confirmed and that it represents a missense variant in a different isoform. In family F2516, a homozygous *IQSEC1*:NM_001134382.3:c.2978del;p.(Pro993Hisfs*127) variant was identified in the index patient but also the normal father and another individual in our database who lacks the phenotype. The deleted bp is the last bp in the exon followed by a single bp intron raising the possibility of a resulting in-frame deletion rather than a frameshift. Another example is *NPHP4*:NM_015102.5:c.2818-2 T > A, with an extremely high AF in gnomAD including homozygotes. The lack of phenotype could be related to the in-frame nature of the splicing aberration[28]. Similarly, *OFD1*:NM_003611.3:c.2600-1 G > C, which we identified in hemizygosity in 3 families was found to cause in-frame aberration when we tested it on RT-PCR (Supplementary Figure 3). *EVC*: NM_001306090.2:c.2731 C > T;p.(Arg911*) is a variant that has a very high frequency in our population with several homozygotes that lack the expected ciliopathy phenotype. We think it is not LOF because it only truncates 8% of the protein. Similarly, *RP1L1*: NM_178857.6:c.5959 C > T;p.(Gln1987*) variant with a very high frequency is probably non-pathogenic because it only truncates a small part of the C-terminus that has not been reported to harbor any pathogenic variants. Furthermore, the truncated part resulting from *GNAT1*:NM_144499.3:c.858 C > G;p.(Tyr286*) we identified in family F5037 with horizontal gaze palsy and in a normal parent, is devoid of clinically proven pathogenic variants.

3. The reported gene-disease assertion is refuted: *LIPN*:NM_001102469.2:c.302delG;p.(Gly101Glufs*7) was identified in homozygosity in F2178 in the absence of ichthyosis. The OMIM listing of *LIPN*-related ichthyosis is based on a single family.

 i. Presumptive LOF variants in genes with no OMIM phenotypes: Although priority was always given to known disease-related genes, the finding of homozygous truncating variants in novel genes can distract from the actual disease-causing variant especially if the novel gene is compelling. Supplementary Data 15 lists all the "knockout" events that turned out to be unrelated to the phenotype in question and these were observed in 60 families (3.8%).

 ii. The variant was in linkage disequilibrium with the causal variant, which was therefore overshadowed by the other variant. For example, family F168 was referred to us with epidermolysis bullosa. We initially made a genetic diagnosis based on *LAMB3*:NM_000228.3:c.2723 C > T;p.(Thr908Ile). However, the variant was subsequently found to have a high AF, which prompted us to reclassify this variant and revisit the case. This revealed a pathogenic NM_000228.3:c.958_1034dup;p.(Asn345-Lysfs*77) variant in the same gene. Similarly, in family F4429 with congenital adrenal hyperplasia we made a molecular diagnosis based on *CYP21A2*:NM_000500.9:c.92 C > T;p.(Pro31Leu) until we later discovered the causal variant to be a genomic rearrangement involving exon 1-3 of the same gene. In family F5113, WGS reported a very deep intronic variant NM_003477.2:c.1023+2267 A > G in *PDHX* as the likely cause. However, the causal variant was genomic rearrangement resulting in exon 1 deletion of the same gene as confirmed by targeted analysis.

 i. Technical:
    a. Deep intronic variants: Supplementary Data 16 for tentative TDV (see above) includes variants >50 bp from the exon-intron junction, which is typically not covered by exome capture.

 b. Regulatory elements (Supplementary Data 17): We highlight a remarkable example of this challenging class. Family F3029 comprises four children with severe syndactyly (Fig. 4a–f). We initially reported a large homozygous genomic deletion NC_000010.10: g.54337730_54933961del as the likely causal variant and attributed the pathogenesis to the resulting total loss of *MBL2* within the deleted region even though *MBL2* is not linked to any disease in humans[29]. Interestingly, a mouse model homozygous for a null allele[30] did not corroborate the phenotype observed in the four siblings prompting us to revisit our interpretation. Upon further investigation, we found that *DKK1* lies ~260 kb upstream of the deleted region (Fig. 4g). DKK1 is crucial for normal limb development and null mice display syndactyly[31] (Fig. 4h). We hypothesized that the deletion may have impacted *DKK1* transcription indirectly by effecting a putative enhancer region. Indeed, we observed several H3K27Ac peaks in the deleted genomic region when inspecting publicly available databases (Fig. 4g). Consistent with this hypothesis, RT-qPCR experiments using fibroblasts isolated from two affected siblings showed a marked reduction (60-80%) of *DKK1* transcript levels compared to controls (Fig. 4i).

 c. Repeat expansion (Supplementary Data 18): As expected, all families with Fragile X syndrome and other expansion disorders were missed by exome except for one case caused by a de novo indel in *FMR1*[32].

 d. Genomic rearrangements: Variants >50 bp in size but below the limit of detection of chromosomal microarray were challenging to call on exome sequencing. These accounted for the molecular diagnosis in 42 families (2.7%, Supplementary Data 19). Supplementary Fig. 4 shows examples of how optical genome mapping was very helpful in this class of variants.

 e. Pseudogenes: The known limitation related to *SMN1*/*SMN2* and *CYP21A2* loci was encountered in 13 families (0.8%).

 f. Platform and bioinformatic limitations: We have encountered 68 families (4.3%) in which the causal variant was missed or miscalled (sometimes because of discrepant performance of Ion Proton vs Novaseq, especially for indels). We have instances where the variant was misannotated as intronic or as ncRNA, but it was actually in an exon of a protein-coding gene.

 g. Epigenetic: These epigenetic changes are missed by current short read sequencing technology e.g. hypomethylation of the maternal *GNAS* allele-related Pseudohypoparathyroidism Ia in family F79, and DMR2 hypomethylation-related Beckwith-Wiedemann syndrome in family F8188.

4. Pedigree-related
 Pseudodominance was encountered in 15 families (~1%, Supplementary Data 20). Examples of more challenging pedigrees are discussed here. The index in family F2640 was suspected to have an autosomal recessive syndromic form of intellectual disability because of the consanguineous parents and history of a deceased affected sibling so a novel candidate gene (*FAM120AOS*) was proposed[33]. However, reanalysis revealed a causal 179 kb deletion that had been dismissed because it was inherited from a normal mother. This deletion (chr14:101178072-101457155) spans an imprinted locus with the paternally expressed *RTL1* and maternally expressed *MEG3* and *MEG8*, and is linked to Temple and Kagami-Ogata syndrome[34] (Fig. 2). F6386 family was referred to us with two affected sisters diagnosed with hearing impairment, mild microcephaly, developmental delay, and mild strabismus. Despite the absence of consanguinity between the two parents, the hearing impairment phenotype was solved with a homozygous truncating variant in *GJB2*:NM_004004.6:c.35delG;p.(Gly12-Valfs*2). To our surprise, we found a heterozygous variant

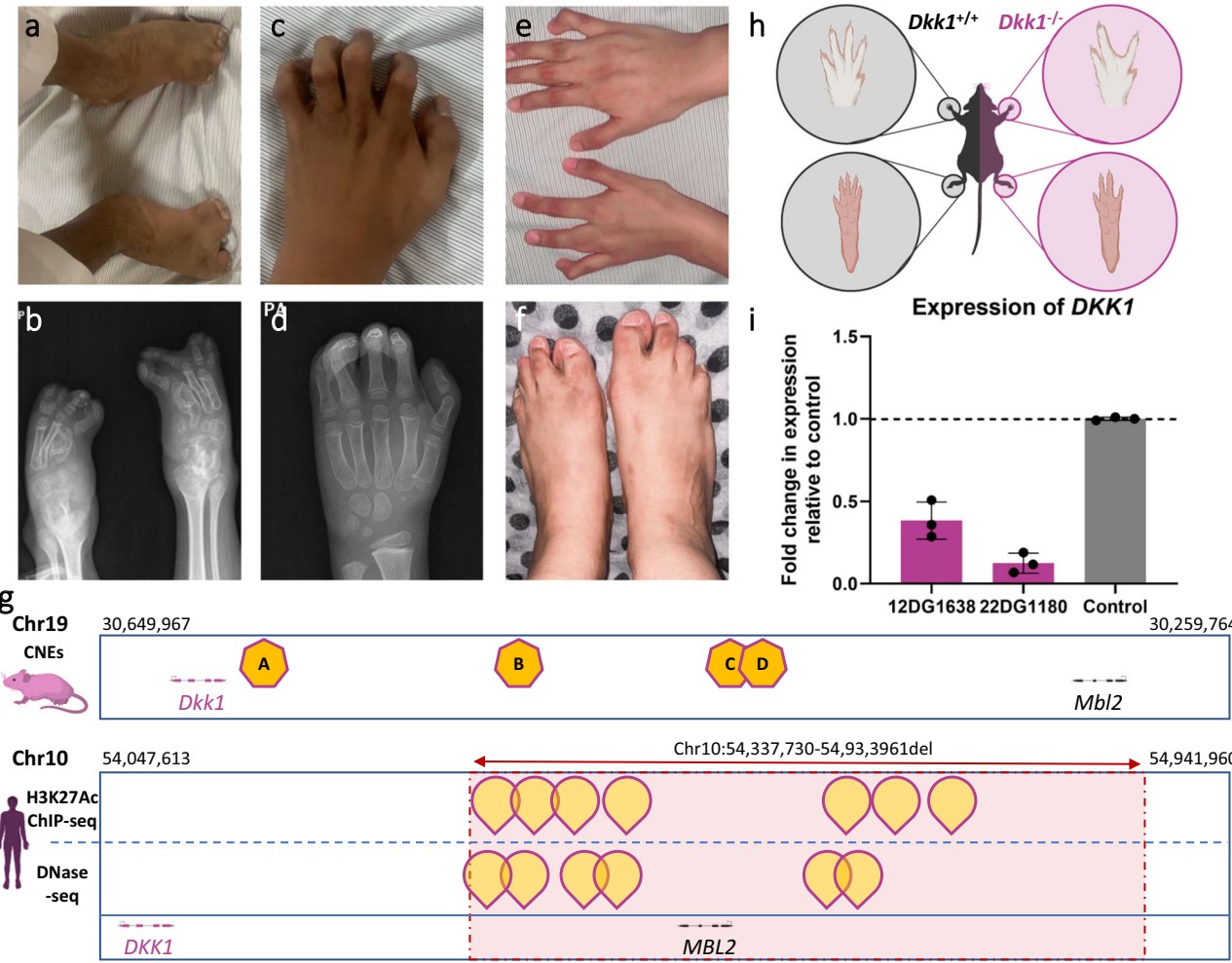

**Fig. 4 | Identification of a homozygous deletion affecting the regulatory elements of *DKK1* in four siblings with complex syndactyly. a** Clinical image of index individual of family F3029 showing syndactyly in feet. **b** X-ray images of the feet. **c** Photograph of the hands showing syndactyly. **d** X-ray of the hands. **e** Clinical image of the hands of a similarly affected sister with syndactyly of the middle finger. **f** Clinical image of the feet of the affected sister with syndactyly. **g** Representative illustration of *DKK1* gene in mouse and human genomes. In mouse, the highlighted regions A, B, C, and D correspond to the four conserved non-coding elements (CNE25, CNE114, CNE190, and CNE195, respectively) identified downstream of *DKK1* and are shown to drive its expression[62]. In humans, peaks corresponding to H3K27Ac and DNase-seq experiments are also highlighted in the deleted region identified in the affected members of this family. **h** Schematic representation of WT and *Dkk1* knockout mice showing syndactyly in hands and feet. **i** RT-qPCR experiment measuring the transcript levels of *DKK1* in the index individual and his affected sister compared to the control. Data show 60–80% reduction in *DKK1* transcript levels. Data are presented as mean values +/− standard deviation (SD). Error bars represent the SD of three experiments. Panels (**h**) and (**g**) are created with BioRender.com.

*ITPR1*:NM_001378452.1:c.7660 G > A;p.(Gly2554Arg) in both affected sisters (explaining the global developmental delay phenotype) which was absent in the parents strongly suggesting a gonadal mosaic mode of inheritance. Another dramatic example is family F4923 with three affected children from two different half second-degree unions with split hand/foot malformation syndrome (Fig. 5a, b). Molecular karyotyping was initially negative as were exome and RNA-seq. Optical genome mapping revealed a heterozygous duplication in chromosome 10 (chr10:102909908-103459101) in all three affected while the mothers and father were wildtypes indicating a case of paternal gonadal mosaicism (Fig. 5c–e). Gonadal mosaicism is even more challenging in the context of autosomal recessive diseases. For example, in family F8654 with two children with the classical *ALG3*-related Congenital disorder of glycosylation, type Id we identified a paternally inherited variant NM_005787.5:c.512 G > A;p.(Arg171Gln) and a "de novo" variant in trans, indicating maternal gonadal mosaicism. Family F5162 exemplifies how the challenge of intrafamilial genetic heterogeneity is amplified when the phenotype of the two conditions is similar (Fig. 5f). The three affected siblings were referred to us with intellectual disability, and limited jaw openings (trismus) of variable severity. The family was solved with a homozygous LOF variant in *THUMPD1*:NM_017736.5:c.706 C > T;p.(Gln236*) identified in two of the three siblings. The third sibling was found to have de novo variant in *HIST1H1E*:NM_005321.3:c.265delA;p.(Ser89-Alafs*140) (Fig. 5g–j). Overall, recurrence caused by parental balanced rearrangement or gonadal mosaicism was erroneously assumed to indicate autosomal recessive variants thus delaying the molecular diagnosis and these were observed in 2 (0.13%) and 7 families (0.45%), respectively. We also note the delay in identifying compound heterozygous variants in 16 consanguineous families (1%) because homozygosity was assumed during analysis. Similarly, blended phenotypes are often assumed to be autosomal recessive in consanguineous

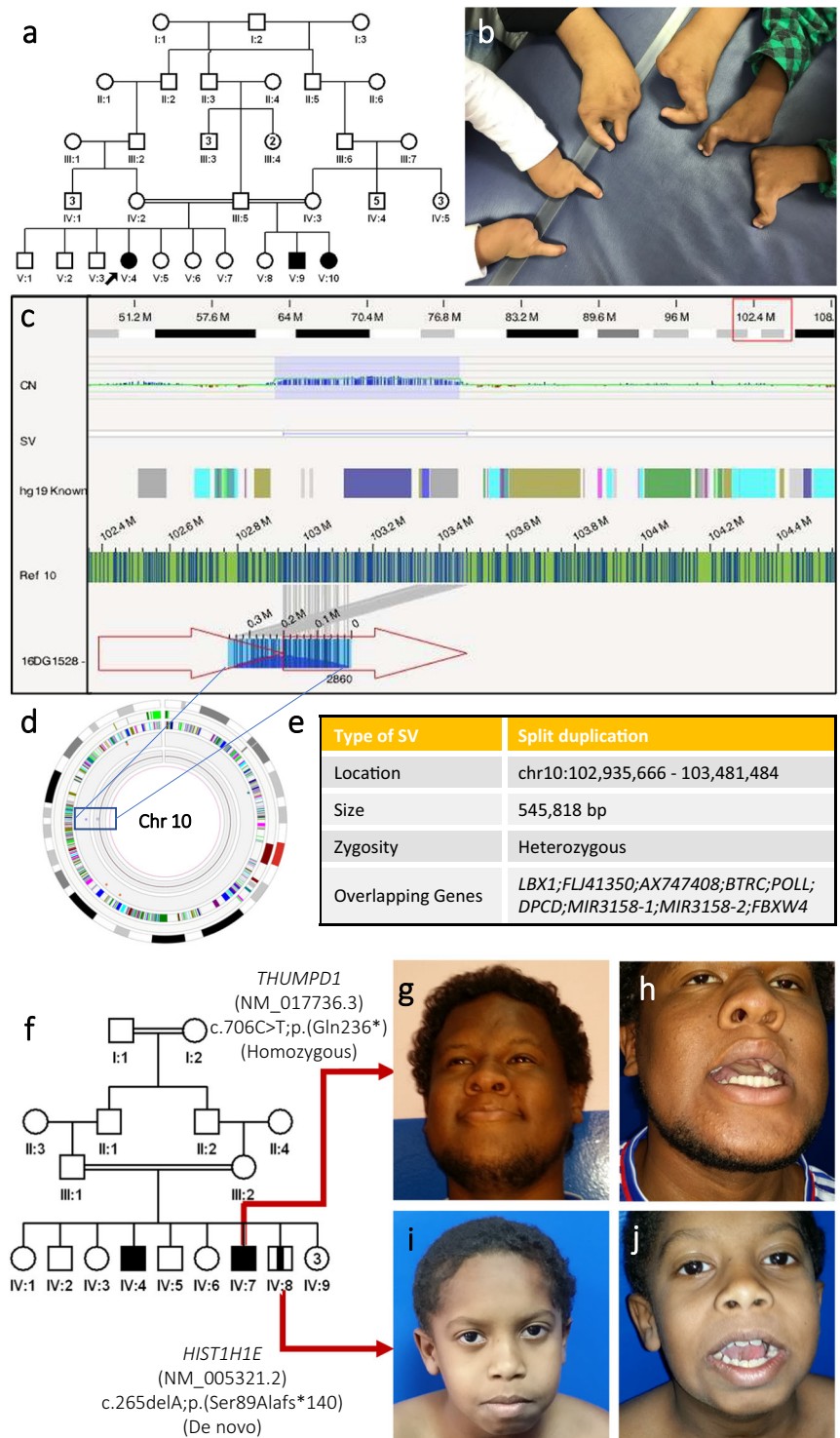

**Fig. 5 | Gonadal mosaicism and intrafamilial genetic heterogeneity are examples of pedigree-related challenges. a** Pedigree of family F4923 with 3 half siblings affected with Split hand/foot malformation syndrome. **b** Clinical image of the three affected siblings highlighting the hand malformations. **c** Screenshot of Bionano analysis output showing the identified heterozygous duplication on Chr10 previously associated with split hand/foot malformation syndrome. **d** Overview of structural variants identified in Chr10 by optical genome mapping. **e** Table summarizing the identified disease-causing lesion in the family. **f** Pedigree of family F5162 with three siblings affected with variable degrees of intellectual disability, abnormal facial shape, and inability to completely open their jaws. Molecular analysis revealed that two siblings were homozygous for a LOF variant in *THUMPD1* while the younger affected sibling had a de novo variant in *HIST1H1E*. **g, h** clinical images of affected individual IV:7 highlighting his inability to completely open his mouth. **i, j** Facial images of affected individual IV:8 demonstrating his incapacity to open his jaw.

populations, but we note that even in individuals with an autosomal recessive allele, the additional pathogenic variant was autosomal dominant in 17.7% or X-linked in 16.1%. Intrafamilial genetic and allelic heterogeneity were major challenges. The default assumption of genetic and allelic homogeneity for a given phenotype within families proved erroneous in 140 families (8.9%), including 123 nuclear families. Supplementary Data 20 summarizes these families, including 70 unpublished ones. This list also includes 87 families in which at least one individual had a blended phenotype (see above). Supplementary Fig. 5 showcases a few remarkable families with genetic and allelic heterogeneity.

5. Positional mapping-related limitations

There were several instances where we excluded a causal variant because it was present within a region of homozygosity (ROH) that was apparently shared with unaffected family members. Notwithstanding the possibility of penetrance (see above), this phenomenon can be explained by IBS (identical by state) rather than IBD (identical by descent). In consanguineous settings, ROH is the surrogate of autozygosity (IBD). However, ROH may also represent IBS. Thus, the apparent sharing of ROH with the unaffected can be due to the region being IBS in the unaffected while IBD in the affected. Typically, IBS is short so it can be very challenging when IBS is long. The largest IBS we have encountered so far is 19.479 Mb long (family F873). This mechanism was encountered in 5 families (0.32%). Conversely, a variant may be dismissed because it is not within an ROH that is shared by all affected. Notwithstanding the possibility of genetic heterogeneity, this phenomenon can be due to double recombination. This is very rare, and we have encountered it in only 2 families. An example is family F849 where the two siblings with Cutis laxa are homozygous for *ATP6V1E1*:NM_001696.4:c.634 C > T;p.(Arg212Trp). The variant was present at the edge of ROH in one affected child but there was no ROH in the affected sibling. We hypothesize that the ROH was abrogated by a second recombination event (Supplementary Fig. 6). The *RP1* locus seems to be particularly susceptible to this phenomenon as we observed it in 3 families where at least one affected member did not have a detectable ROH around the causal variant. Finally, the variant may not be within ROH at all (29 families, 1.9%), and this can be due to low SNP coverage near the locus or because the parents are sufficiently removed from the common ancestor that the disease haplotype was reduced below the conventional limit of 2 Mb. Supplementary Data 21 and Supplementary Fig. 6 summarize positional mapping-related challenges.

7. Sample mix-up:

Despite having multiple checks to avoid human errors, sample mix-up was responsible for a delay in identifying the likely causal variant in 6 families (0.38%).

### Reanalysis of negative cases

In our cohort, 314 families were referred to us after a negative exome, genome, or both. Reanalysis identified a likely causal variant in 54.5% of these families. This offered us an opportunity to explore the relative contribution of the above-described challenges in this special cohort. The single most common challenge was the novelty of the gene-disease assertion (48%, this includes novel disease genes as well as known disease genes with novel mutation mechanism). This was followed by variant-related challenges (genomic rearrangements and non-canonical tentative TDVs) (37.4%), phenotype-related issues (11.7%), and pedigree-related challenges (1.8%). Only 15.2% of the variants identified on reanalysis could not have been captured at the technical level by exome sequencing. Figure 1 and Supplementary Data 22 summarize the results.

## Discussion

NHGRI (National Human Genome Research Institute) has made "bold predictions" for the state of human genomics by the year 2030[35]. One such prediction is that "The regular use of genomic information will have transitioned from boutique to mainstream in all clinical settings, making genomic testing as routine as complete blood counts (CBCs)". Key to this prediction is our ability to interpret genome sequence data. Nowhere in the field of genomics does this interpretation have the potential to be more accurate and attainable than in Mendelian diseases and yet patients suffering from these diseases have at least 50% chance of remaining undiagnosed after clinical genome sequencing. Clearly, this must change to fulfill the above vision and deliver to these patients their right to an accurate diagnosis.

This study is a step towards shedding light on the factors that render genome sequencing non-diagnostic through deep analysis of real-world data from a large Mendelian program with excellent sampling representation from one of the largest countries in the Middle East. Contrary to the common belief that the missing diagnostic yield of exome is mostly related to technical limitations[36,37], we have previously shown using an unbiased positional mapping approach that at least in the setting of autosomal recessive phenotypes in consanguineous populations, more than 90% of causal variants should in theory be detectable by exome sequencing[5]. Indeed, we and others have shown that reanalysis of "negative" exome sequencing uncovers causal variants that were missed at the interpretation rather than capture stages[38–42]. While our study offers limited insight into the added value of newer technologies such as optical genome mapping, it provides unprecedented details about the interpretation challenges.

There is a growing interest in the use of artificial intelligence (AI) to improve accuracy and increase the throughput of interpreting clinical genome sequencing[42]. We believe that efforts such as ours to share challenges in analyzing genomes and how such challenges were overcome will be very helpful in training the next generation of AI-based tools and enable genome sequencing and its interpretation and reporting to be performed at scale. Additionally, our work makes several key contributions to clinical genomics. First, we describe 357 gene-disease assertions that were novel at the time of analysis and provide additional support to 120 previously published tentative gene-disease assertions. This was largely enabled by the power of autozygosity to produce compelling homozygous loss of function variants (human knockouts) as described before[20,43–45]. This is readily seen in the case of *ABL1* homozygous loss of function variant that we suggest produces a mirror-image phenotype to *ABL1*-related Van den Ende Gupta syndrome-like facial and digit dysmorphism caused by de novo gain of function variants. Similarly, the homozygous splicing variant in *OTX2* was shown in our study to result in a human phenotype that recapitulates the retinal degeneration and infertility observed in homozygous mice rather than the haploinsufficiency-related microphthalmia phenotype[46,47]. Beyond these novel disease-gene assertions, we have also added novel aspects to the phenotype of 73 established gene-disease links, which will aid in the molecular diagnosis of these diseases. Second, we publish recessive forms of 23 genes that have hitherto been linked to autosomal dominant phenotypes only. The importance of this finding in interpreting heterozygous pathogenic variants in these genes in individuals who lack the dominant phenotype cannot be overemphasized. This makes the difference between counseling based on a 50% risk of an affected child assuming non-penetrance of a dominant variant versus a nearly 0% risk of an affected child assuming recessiveness of the variant and a non-carrier spouse[48]. Third, we highlight 85 variants as important founder variants in our population including 48 variants with local allele frequency >0.001 (Supplementary Data 23). Importantly, ~50% of these variants have a gnomAD frequency of 0, which highlights their specific importance to the local and potentially other Middle Eastern populations as shown recently[49]. We believe this is an important step towards realizing the

NHGRI prediction that by the year 2030 "individuals from ancestrally diverse backgrounds will benefit equitably from advances in human genomics". Fourth, we provide an interpretation framework for variants in 58 genes that we note harbor homozygous loss of function class variants with no apparent consequences phenotypically. None of these genes had been linked to any Mendelian disease in humans so our study reduces their relevance as candidate disease genes in future studies especially considering that many have corresponding mouse knockout lines with no major developmental consequences. On the other hand, the examples of "human knockouts" we encountered for genes with established links to otherwise completely penetrant Mendelian diseases raises interesting questions about the concept of "resilience". Such cases are extremely valuable to analyze to gain insights into how they remain disease-free and what that can teach us about human genomics-inspired therapies for patients who suffer from the corresponding diseases[50]. Another advantage offered by the enhanced autozygosity in our study population is the ability to observe in the homozygous state known pathogenic recessive variants and the potential discordance of their phenotypic expression in the homozygous vs. compound heterozygous state. The frequency of many of these variants is low enough that it is virtually impossible to encounter them in the homozygous state in outbred populations where the required cohort size is based on $q^2$. A good example of this is *DHCR7*:NM_001360.3:c.1 A > G;p.? with AF of 0.00001 in gnomAD (0 homozygotes), whereas 2 homozygotes were encountered in our much smaller database of <14,000 local exomes. Neither of the two homozygotes displayed features of Smith-Lemli-Opitz syndrome, which lends support to the hypothesis that this variant is only pathogenic in trans with a more severe allele i.e. complex compound heterozygosity.

The small contribution of "technical" challenges we identified in this large cohort is consistent with prior work demonstrating, using an unbiased positional mapping approach, that the overwhelming majority of disease-causing variants, at least in the context of autosomal recessive diseases in consanguineous families, are identifiable by exome sequencing[5,38]. Our work highlights the need to have a thorough approach to cases that remain undiagnosed after clinical exome/genome sequencing by considering all classes of challenges and not focusing solely on improved sequencing technologies. Indeed, many studies on the reanalysis of existing exome and even genome data have demonstrated improved diagnostic rate[39,51,52]. This study suggests that addressing non-sequencing-based challenges alone could boost the diagnostic yield by ~71%. Another important aspect worth highlighting is the value of data sharing and collaboration, which is the motive behind this work. We note that the majority of our novel gene-disease assertions were corroborated by international collaborations we participated in through data sharing (Supplementary Data 24).

In conclusion, we report the largest comprehensive analysis to date on challenges in the identification of causal variants in patients undergoing genome sequencing. Our data argue that investment into new sequencing technologies should be accompanied by similar investment into improved interpretation pipelines if we are to reap the full benefits of clinical genomics. We hope the lessons learned from our analysis will assist the development of such tools and improve the interpretation of genome sequencing both at the variant and the gene levels.

## Methods

We confirm that our research complies with all relevant ethical regulations at KFSHRC.

### Human subjects

Only patients with a suspected Mendelian disease are included in the Mendelian genomics cohort. Informed consent is obtained prior to enrollment. Different IRB-approved projects were used to enroll subjects depending on their phenotype (KFSHRC RAC# 2070023, 2080006, 2121053, 2170028, 2200030, 2080033, 2210029, 2140016, 2230016, and 2080051). The consent covers the use of human fetal material, and for the generation and use of patient-derived cell-lines (LCLs and Fibroblasts) whenever it is needed. The authors affirm that human research participants provided written informed consent for publication of the images in Figs. 2, 3, 4, 5 and Supplementary Fig. 3. The authors also affirm that human research participants provided written informed consent for the publication of identifiable data.

### Testing strategies

Cases recruited prior to 2012 were analyzed using next-generation-based multi-gene panels relevant to their clinical phenotype. Negative cases as well as cases recruited after 2012 were submitted for ES. Simplex cases as well as cases with likely dominant inheritance were additionally subjected to chromosomal microarray. Select negative cases after ES were analyzed using optical genome mapping. All families were submitted for genotyping and positional mapping was used as appropriate. Miscellaneous testing strategies (methylation analysis, MLPA, repeat expansion, and WGS) were requested clinically as appropriate by the ordering physicians from various CAP-accredited laboratories and their results were recorded. Supplemental Methods explain the technical details of the testing platforms.

### Cell culture and RT-qPCR

A sub-confluent patient- and control-derived fibroblast cell lines were maintained in Minimum Essential Medium Eagle (MEM) supplemented with 10% fetal bovine serum (FBS, Gibco), 2 mM L-glutamine (Gibco) and penicillin G (200 U ml-1, Gibco). Lymphoblastoid cell lines were maintained using RPMI media supplemented with 15% fetal bovine serum (FBS, Gibco), 1% L-glutamine (Gibco), and 1% penicillin G (200 U ml-1, Gibco). Cell lines were maintained in a humidified 37 °C incubator at 5% $CO_2$. For RT-PCR and RT-qPCR total RNA from either fibroblast or lymphoblastoid cell lines were extracted with the QIAamp RNA Mini Kit (QIAGEN) according to the manufacturer's recommendations. Preparation of the cDNA was carried out with the iScriptTM cDNA synthesis kit and Poly T oligonucleotide primers (Applied Biosystems). *GAPDH* was used as an internal control. Relative quantitative RT-PCR for expression analysis was performed with SYBR Green and Applied Biosystems StepOnePlus Real-Time PCR System (StepOneTm Software).

### Variant annotation and classification

Candidate variants and their familial segregation were confirmed using Sanger sequencing. All variants are listed using MANE Select or MANE Plus Clinical when applicable and classified according to ACMG guidelines[53] with the help of VARSOME[54], which was subsequently verified manually. Segregation level support of variant pathogenicity ranged from supporting to strong as detailed in[55]. The tool ConsCal was used to increase the throughput of segregation level analysis[56]. Novel genes and known genes with novel allelic disorders were classified using ClinGen gene-disease assertion guidelines[57]. Local allele frequency (AF) was calculated based on an in-house dataset of 13,473 local exomes.

### Review of challenging scenarios

A detailed review of every single molecularly characterized family was undertaken to identify instances that are deemed challenging by a standard approach[58]. Categorization of the challenges followed a consensus approach among the scientific team. All families that were susceptible to a given challenge were counted.

**Reporting summary**

Further information on research design is available in the Nature Portfolio Reporting Summary linked to this article.

## Data availability

All data (including Sanger sequencing, Optical genome mapping, and molecular karyotyping) supporting the findings described in this manuscript are available in the article, its Supplementary data files, and from the corresponding author upon request. Due to local privacy laws and privileged human information, all requests for WES and WGS data are subject to prior approval from the local IRB. Local IRB can be reached at ORA@kfshrc.edu.sa with an expected timeframe for response of two months. All variants have been submitted to ClinVar (https://www.ncbi.nlm.nih.gov/clinvar/).

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

## Acknowledgements

We thank the families for their enthusiastic participation. We acknowledge the genotyping and sequencing support from the Center for Genomic Medicine. We thank Drs. Ahmed A Alhumidi and Asim Alrumaih for their help with Supplemental Fig. 3g. This study was partially supported by The Bill-Melinda Gates Foundation (237835), National Institute of Health (239085), and the Intramural Research Fund for an extended project to study the common genetic diseases in Saudi Arabia [Demography of Recessive Diseases in KSA; Grant #019-052].

## Author contributions

F.S.A., L.A., and S.M. contributed to study conception and design. Data collection and analysis was performed by: L.A., S.M., H.E.S., E.K., R.H., H.H., A.A., M.O.H., F.Abdulwahab, O.A., M.A., N.A., A.J., T.A., F.Alzahrani, Mendeliome Group, and FSA. F.S.A. and L.A. drafted and prepared the manuscript. All authors reviewed the manuscript and approved the final version.

## Competing interests

The authors declare no competing interests.

## Additional information

# Mendeliome Group

Afaf I. Al-Sagheir[5], Ahmad M. Mansour[6], Ali Alawaji[7], Amal Aldhilan[4], Amal Alhashem[4,8,9], Amal Alhemidan[10], Amira Nabil[11], Arif O. Khan[12,13], Aziza Aljohar[14], Badr Alsaleem[15], Brahim Tabarki[4,16], Charles Marques Lourenco[17,18], Eissa Faqeih[19], Essam AlShail[20], Fatima Almesaifri[21], Fuad Al Mutairi[22,23], Hamad Alzaidan[24], Heba Morsy[11,25], Hind Alshihry[26], Hisham Alkuraya[27], Katta Mohan Girisha[28,29], Khawla Al-Fayez[4], Khalid Al-Rubeaan[30], Lilia kraoua[31,32], Maha Alnemer[8,33], Maha Tulbah[33], Maha S. Zaki[34], Majid Alfadhel[22,35], Mohammed Abouelhoda[36], Marjan M. Nezarati[37], Mohammad Al-Qattan[38], Mohammad Shboul[39], Mohammed Abanemai[5], Mohammad A. Al-Muhaizea[8,20], Mohammed Al-owain[24], Mohammed Sameer Bafaqeeh[40], Muneera Alshammari[41], Musaad Abukhalid[20], Nada Alsahan[33], Nada Derar[24,42], Neama Meriki[43,44], Saeed A. Bohlega[20], Saeed Al Tala[45], Saad Alhassan[33], Sami Wali[4], Sarar Mohamed[4,8,46], Serdar Coskun[8,47], Sermin Saadeh[5], Tinatin Tkemaladze[48], Wesam Kurdi[8,33], Zainab Ahmed Alhumaidi[49] & Zuhair Rahbeeni[24]

[5]Department of Pediatrics, King Faisal Specialist Hospital and Research Center, Riyadh, Saudi Arabia. [6]Department of Ophthalmology, American University of Beirut, Beirut, Lebanon. [7]Genetic Center, Prince Mohammed bin Nasser Hospital, Jazan, Saudi Arabia. [8]College of Medicine, Alfaisal University, Riyadh, Saudi Arabia. [9]Sehha Virtual Hospital, Ministry of Health, Riyadh, Saudi Arabia. [10]Department of Ophthalmology, Prince Sultan Medical Military City, Riyadh, Saudi Arabia. [11]Human Genetics Department, Medical Research Institute, Alexandria University, Alexandria, Egypt. [12]Eye Institute, Cleveland Clinic, Abu Dhabi, UAE. [13]Department of Ophthalmology, Cleveland Clinic Lerner College of Medicine of Case Western Reserve University, Cleveland, OH, USA. [14]Department of Dentistry, King Faisal Specialist Hospital and Research Center, Riyadh, Saudi Arabia. [15]Gastrointestinal Division, Intestinal Failure Program, Children's Specialized Hospital, King Fahad Medical City, Riyadh, Saudi Arabia. [16]Pediatric Neurology, University of Sousse, Sousse, Tunisia. [17]Faculdade de Medicina de São José do Rio Preto (FAMERP), São Jose do Rio Preto, São Paulo, Brazil. [18]Specialized Education at DLE/Grupo Pardini, in Personalized Medicine área, Belo Horizonte, MG, Brazil. [19]Section of Medical Genetics, Children's Specialized Hospital, King Fahad medical City, Riyadh, Saudi Arabia. [20]Neuroscience Centre, King Faisal Specialist Hospital and Research Center, Riyadh, Saudi Arabia. [21]Pediatric Department, Clinical Genetic and Metabolic at Hamad Medical Corporation, Doha, Qatar. [22]Department of Genetics and Precision Medicine, King Abdullah Specialized Children Hospital, King Abdulaziz Medical City, Ministry of National Guard Health Affairs, Riyadh, Saudi Arabia. [23]King Saud bin Abdulaziz University for Health Sciences, King Abdullah International Medical Research Centre, Ministry of National Guard Health Affairs, Riyadh, Saudi Arabia. [24]Department of Medical Genomics, Centre for Genomic Medicine, King Faisal Specialist Hospital and Research Center, Riyadh, Saudi Arabia. [25]Department of Neuromuscular Disorders, UCL Institute of Neurology, University College London, Queen Square, London, United Kingdom. [26]Department of Dermatology and Dermatosurgery, Prince Sultan Medical Military City, Riyadh, Saudi Arabia. [27]Global Eye Care, Specialized Medical Center Hospital, Riyadh, Saudi Arabia. [28]Department of Medical Genetics, Kasturba Medical College, Manipal, Manipal Academy of Higher Education, Manipal, India. [29]Department of Genetics, College of Medicine & Health Sciences, Sultan Qaboos University, Muscat, Oman. [30]Research and Scientific Center, Sultan Bin Abdulaziz Humanitarian City, Riyadh, Saudi Arabia. [31]Department of Congenital and Hereditary Diseases, Charles Nicolle Hospital, Tunis, Tunisia. [32]LR99ES10 Human Genetics Laboratory, Faculty of Medicine of Tunis, University of Tunis El Manar, Tunis, Tunisia. [33]Department of Obstetrics and Gynecology, King Faisal Specialist Hospital and Research Center, Riyadh, Saudi Arabia. [34]Clinical Genetics Department, Human Genetics and Genome Research Institute, National Research Centre, Cairo, Egypt. [35]Medical Genomic Research Department, King Abdullah International Medical Research Center, King Saud Bin Abdulaziz University for Health Sciences, Ministry of National Guard Health Affairs, Riyadh, Saudi Arabia. [36]Department of Computational Science, Center for Genomic Medicine, King Faisal Specialist Hospital and Research Center, Riyadh, Saudi Arabia. [37]Department of Genetics, North York General Hospital, University of Toronto, Toronto, ON, Canada. [38]Department of Surgery, College of Medicine, King Saud University, Riyadh, Saudi Arabia. [39]Department of Medical Laboratory Sciences, Jordan University of Science and Technology, Irbid, Jordan. [40]Pediatric Nephrology Department, Children's Hospital, King Saud Medical City, Riyadh, Saudi Arabia. [41]Pediatric Department, King Saud University Medical City, King Saud University, Riyadh, Saudi Arabia. [42]Genetics Department, Yale University School of Medicine, New Haven, CT, USA. [43]Department of Obstetrics and Gynecology, College of Medicine, King Saud University, Riyadh, Saudi Arabia. [44]Maternal Fetal Medicine, King Khalid University Hospital, Riyadh, Saudi Arabia. [45]Department of Pediatrics, Armed Forces Hospital, Khamis Mushayt, Saudi Arabia. [46]Department of Pediatrics, Faculty of Medicine, National University, Khartoum, Sudan. [47]Department of Pathology and Laboratory Medicine, King Faisal Specialist Hospital and Research Center, Riyadh, Saudi Arabia. [48]Department of Molecular and Medical Genetics, Tbilisi State Medical University, Tbilisi, Georgia. [49]Pediatric & Genetic Department, MCH Dammam, Ministry of Health, Dammam, Saudi Arabia.

