## [Peer Review File · Nature Communications]

The Diagnostic Implications of Pitfalls in Causal Variant Identification Based on 4,577 Molecularly Characterized FamiliesREVIEWER COMMENTS

Reviewer #1 (Remarks to the Author):

AlAbdi et al provide a thorough review of the diagnostic challenges they encountered when evaluating more than 4000 mainly consanguineous families with rare genetic diseases. They propose a systematic approach to classify these challenges into categories and then highlight that the main limitations to identify a diagnosis are caused by interpretation issues rather than technical issues. These issues have been reported before but to the best of my knowledge, there has not been a cohesive reporting of the prevalence of these types of features in analysis across a large cohort.

The novel disease gene associations reported do not meet the standards required in the field for the number of cases required or the level of phenotype description, or the often needed functional evaluation.

This article will be of interest to the clinical genetics community as it highlights considerations in genomic analysis including the importance of careful phenotyping. Overall, the manuscript reads like a long list which may not be appealing to some readers. Please consider the following points in the revision process:

Major comments

1. Study cohort description – would be helpful to include the sequencing modalities applied to this cohort (exome, genome, etc) and sequencers (as there is later discussion of analysis differences due to different platform use).
2. Category 1 i): In the F5711 family description, there is mention of the father's facial dysmorphism but it is not mentioned if this is present or absent in the children (assume absent because not mentioned but better to state what is and is not present given the focus here is phenotype).
3. Category 1 iv): "Hybrid phenotypes" does not clearly capture what is included in that category in my opinion. Multilocus phenotypes could also suggest a polygenic inheritance rather than monogenic conditions. The words "blended phenotypes", "dual diagnoses" or "multiple genetic diagnoses" could be considered to name that category.
4. Category 2 i) Reporting 1-2 families with interesting variants in a gene not yet associated with disease does not make a novel gene-disease association. The field standard for publishing these is typically at least 3 cases with extensive phenotypic and molecular description (though case reports are sometimes published typically to raise awareness of a gene to gather more cases to eventually be able to report a novel disease association for a gene). Reporting fewer cases in a table without extensive phenotype data

is an insufficient contribution to the literature and these should not be described as novel disease gene assertions.

5. Category 2 iii) “Distinct mutational mechanism” is a subgroup of novel gene-disease assertions (category 2 i)). A name like “novel gene-disease assertion in a gene already known to cause another condition, through a distinct molecular mechanism” or “known gene, novel mutation mechanism” could be considered or if the authors find another way to link it to category 2i.

6. Category 3 i b 6) “There is no clear reason for the discrepancy between the high MAF and observed disease phenotype.” Two variants are given but the allele frequencies are not reported. Checking gnomAD, the DYNC2H1 variant is in 7 people (https://gnomad.broadinstitute.org/variant/11-102991434-C-T?dataset=gnomad_r2_1) and 1 person in the GME database and the DALRD3 variant in 1 person (https://gnomad.broadinstitute.org/variant/3-49053669-G-T?dataset=gnomad_r2_1). It would be good to mention the AF in the manuscript and also these AFs are not particularly high.

7. The challenges of non-coding RNA genes should be discussed and probably have its own category. These genes can be captured by exome sequencing, but filtered out and not included in the variant files for analysis by many pipelines. The example of RNU4ATAC disorder is only mentioned in the in silico category. Can the authors comment on diagnoses in non-coding RNA genes in their cohort? Several have been linked to human disorders.

8. International data sharing of variants identified in patients with rare genetic diseases is a major contributor to improve variant interpretation, but also a major challenge. Can the authors discuss this and how data sharing is impacting their analysis?

Minor comments

1. Line 255 – synonyms instead of synonymous

2. Line 258 – RT-PCR experiments noted altered splicing. Was this variant of interest because of splice predictors? SpliceAI score is 0.35 for donor gain.

3. Line 260 – “a default MAF of <0.001 is often used as a threshold for rare recessive disease” – this is a very conservative threshold compared to what I normally see reported which is <0.01. Is there a reference for this lower threshold?

4. Line 270 – “The variant is only pathogenic when compound heterozygous” – consider adding “with a more damaging variant” as I think that is what you want to convey here.

5. Line 364 – should be lowercase pLI

6. Line 368-9 – “Another example is RPGR variant nomenclature, which we found” – add the comma and word which

7. Line 423 – capitalize F in Fragile X syndrome

8. Line 458 – use the term exome rather than whole-exome

9. Line 463 – “limited jaw openings to variable severity” to “limited jaw openings of variable severity”

10. Figure 1 bottom left panel categories are unclear from the figure.
11. Figure 5 legend does not have text for parts D and E.
12. In table 2, there is a typo in “mechanism”. Also, “Unusual mutation mechanism” is not clear. A name like “known gene, novel mutation mechanism” or something more descriptive could be considered.
13. Table 4 is busy. For ease of reading, the authors could group together cases having the same justification of non-penetrance into categories “inadequate phenotyping”, “true non-penetrance”, etc in the first column.
14. The abbreviation MAF is not defined in the manuscript though minor allele frequency without abbreviation is listed in the Figure S2 name. The minor allele frequency is the second most common allele in a given population. Given many sites are multiallelic, this is likely not the term the authors mean in all cases and better to use the term “allele frequency”.

Reviewer #2 (Remarks to the Author):

This manuscript by AlAbdi and presents a long list of challenges encountered when analyzing sequencing data from rare disease patients with suspected Mendelian disorders. On the one hand, many of these are challenges that can be encountered and highlighting them could be useful lessons learned. However, there are some major issues with the claims presented that are cause for concern in drawing too many conclusions. I evaluated most of the first 10 or so variants that were called out as causal in the early part of the manuscript, and then a few others here and there and found issues with the majority of them in terms of their qualifications as pathogenic and causal for the phenotype noted. As such, the overall premise of using these cases as exemplars for the challenges noted seems problematic. In general, it does not appear that the authors followed current professional guidelines of evaluating variants for pathogenicity using the 2015 ACMG/AMP guidelines, nor have they ensured that the novel gene-disease associations meet at least moderate by ClinGen standards if they are to consider the findings anything beyond a plausible candidate. I have also identified a few other more specific points as noted below.

TBX1 - NM_080647.1:c.1158_1159delinsT;p.(Gly387Alafs*73) - This variant is in the terminal exon and therefore not expected to undergo NMD. Insufficient evidence of pathogenicity has been presented, especially given the poor match in phenotype for the family, as the authors point out.

INSR - NM_000208.2:c.433C>T;p.(Arg145Cys) – No evidence is provided by authors. There is one zero star entry in ClinVar as Likely Pathogenic but it also has no evidence provided.

TMPRSS15 - NM_002772.2:c.2808_2809insATCA;p.(Ser937Ilefs*4) – submitted to ClinVar by the authors but with no evidence and no publication link; The PMID listed leads to a paper by the authors but with no evidence presented - this is the most C-terminal pLOF variant in ClinVar and its not clear if it occurs after the NMD site – more evidence for pathogenicity needed.

FHR - NM_000145.2:c.1030G>A;p.(Val344Met) – unpublished; present in gnomAD but not in ClinVar; no evidence for pathogenicity demonstrated.

OTX2 - NM_001270525.2:r.98_273del;p.(Pro34Metfs*3) – this variant occurs in a portion of exon 2 that is alternatively spliced out in other transcripts. No other P/LP variants have been observed in this region. Despite gnomAD being highly constrained for pLOF variants (only 1 pLOF), the one present is in this alternately spliced region of exon 2. Further evidence of pathogenicity is needed.

CD151 - NM_001039490.1:c.493C>T;p.(Arg165*) – there are only 2 P/LP variants reported in ClinVar, spanning at most 4 families so its not surprising that the phenotypic spectrum was poorly defined. Indeed, another paper was just published documenting a third variant in a family with renal agenesis (PMID: 35707593).

COL25A1 - NM_001256074.2:c.1517delC;p.(Pro506Hisfs*25) and large deletion involving COL25A1 and ZCCHC23 – The COL25A1 gene has been linked to an AR inherited disorder called congenital fibrosis of extraocular muscles. The authors state the gene is linked to a new AD disorder involving fetal akinesia based on two pLOF variants identified. However, no evidence is presented to reconcile the fact that the gene is unconstrained for LOF and littered with pLOF variants throughout the coding region.

HADHB - NM_000183.2:c.712C>T;p.(Arg238Trp) – This variant was submitted to ClinVar by the authors (as Likely Pathogenic) and by two clinical labs as VUS. Its also present in gnomAD. The authors did not provide any evidence to support pathogenicity.

SCLT1 - NM_144643.2:c.778_780del;p.(Glu260del) – The authors state that this case has BBS due to this variant despite the gene being linked to OFD syndrome. However, the gene is not linked to any disorder in OMIM other than a note that says “This variant is classified as a variant of unknown significance because its contribution to orofacioidigital syndrome IX (258865) has not been confirmed” based on a 2014 publication. The genes have been submitted as Limited for BBS and Senior-Loken syndrome in GenCC. So there does not seem to be sufficient evidence for a link to any disorder including BBS and the authors have not presented any data to support their claim.

CDK10 - NM_052988.3:c.139delG;p.(Glu47Argfs*21) – this one does appear to be pathogenic but it is listed as unpublished but was published by another group in 2017 (PMID: 28886341), is present in OMIM and submitted to ClinVar.

PIK3C2G - NM_004570.5:c.3101G>A;p.(Arg1034His) - This is the first example described for hybrid phenotypes and was found in a patient with a MERTK deletion causing RP. The authors argue that the PIK3C2G variant is responsible for their insulin resistance. However, no evidence has been presented for why this variant was assumed causal. The gene has not been implicated in any diseases in OMIM. A search in ClinVar finds one entry from a research lab proposing it as a candidate gene for Tracheoesophageal fistula and a literature search finds one small non-replicated GWAS study from 15 years ago that finds an association with 2 SNPs in this gene and provides an OR of 2. I do not think this is sufficient evidence to say that this gene or variant has any role in the patient's insulin resistance.

The authors state "The clinical utility of causal variant identification in Mendelian diseases entails the provision of a precise diagnostic label, informing management decisions and empowering individuals (patients and unaffected carriers) to make reproductive choices." I would add "understand disease risk in family members and future generations" as an equally important element of clinical utility.

The authors state "When the underlying molecular diagnosis was considered challenging due to factors unique to a particular family but not others, only that family counted towards the tally of challenging cases. Otherwise, all families that were susceptible to a given challenge were counted." This is confusing as I'm trying to figure out why they can't just state "All families that were susceptible to a given challenge were counted." Is there something about the first sentence that makes the second sentence on its own inaccurate?

The authors include a method for OPT but it seems most cases were analyzed by WES. Please include all approaches in the methods.

Please include a better description of the cohort including where cases were recruited from. Were they all research subjects or were some cases from clinical testing services? What was the race/ethnicity/ancestry of the population? Was it enriched for consanguinity?

For category 1- Phenotype-related: iii) Allelism, how much evidence is there for these new allelic diseases at the variant and gene level, especially the 41 that are not even entered in OMIM, which has a low bar of evidence for entry. For example, how many of the distinct allelic disorders had sufficient evidence for the gene-disease relationship to reach at least moderate evidence according to ClinGen standards? It would be useful to delineate how many valid new allelic diseases were due to distinct inheritance patterns (e.g. AR vs AD) and/or mutation types (e.g. pLOF vs GOF/misense only).

For the Gene-related section 2- i) on novel gene-disease assertions, the list spans genes which have been subsequently implicated in disease as the authors state, but it also includes genes that are simply candidates based on a single family (e.g. DNAJC17), and even 23 genes that are obscured with a “a” symbol as the authors do not wish to reveal the discovery. Similar “a” annotations are in other tables. Please remove these genes if you are unwilling to identify them. Also, it would be important to distinguish those genes that do not yet have sufficient evidence to be validly implicated in disease and are still candidates. Use of ClinGen’s framework for gene disease validity is suggested. It would also be useful to know how many are in the GenCC database. It seems the senior member of this paper is a submitter to GenCC yet these genes have not yet been submitted. Is there a reason why?

In section 3-i-a, the authors describe all splice region variants as TDV (transcript deleterious variants) even before they have been evaluated. I would suggest describing them as splice region variants and only identify the subset as TDV once proven to impact splicing. Also, please be sure all variants are being evaluated and reported use the MANE Select or MANE Plus Clinical transcripts as the current standard.

In section 3-i-b, the authors highlight variants that were challenging due to high MAF. Yet the majority of the variants in the list are well-established pathogenic variants listed as such in ClinVar. Was a standard filter not used to identify pathogenic variants in ClinVar? Other variants were called out as not yet being recognized as more common rare diseases (e.g. WASHC5 and NT5DC1) yet evidence for pathogenicity was not presented as noted below and each was only seen in one proband (RECQL4 also seen in only one). Please provide evidence for pathogenicity and a common role in disease.

WASHC5 - NM_014846.3:c.2849A>G;p.(Lys950Arg) - No evidence for pathogenicity provided. VUS in ClinVar

NT5DC1 NM_152729.3:c.1114T>C;p.(Ser372Pro) — No evidence for pathogenicity provided.

CEP250 - NM_007186.2:c.1826C>T;p.(Ala609Val) – No evidence for pathogenicity provided. VUS in ClinVar

In the “5- Positional mapping-related limitations” section, the authors describe confusion over the apparent familial segregation discordance related to ROH due to assuming IBD when the ROH was due to IBS. Yet its not clear if they ever looked at the specific variant under question in these situations, which I presume would have shown the accurate segregation with disease if it was causal. While I understand that use of ROH in consanguineous families can be a useful way to focus in on the most likely regions implicated in disease, I would assume that it is only a first pass strategy and not something relied on.

The authors state “Contrary to the common belief that the missing diagnostic yield of exome is mostly related to technical limitations...”. However, I’m not aware that this is a common belief and instead I

think most believe the missing yield is due to our inability to interpret most variation as causal. If the authors wish to state this, please reference sources.

The authors state "... since it makes the difference between counseling them for a 50% recurrence risk on the assumption of non-penetrance to a nearly 0% recurrence risk on the assumption of recessiveness and non-carrier spouse." In most settings of genetic disease counseling, recurrence risk is often focused on the chance of having a second child with the disease, which in this case would be 25%, not 0%. It may be worth adding that point as well.

The authors note that they "... provide an interpretation framework for variants in 52 genes that we note harbor homozygous loss of function class variants with no apparent consequences phenotypically" yet I don't see this list anywhere in the materials provided. Also, have the pLOF variants been evaluated to ensure they are subject to LOF as opposed to escaping NMD or removing the variant's effect through alternative splicing, etc?

Reviewer #1:

We are pleased to read the respected reviewer's positive comments such as "These issues have been reported before but to the best of my knowledge, there has not been a cohesive reporting of the prevalence of these types of features in analysis across a large cohort" and "This article will be of interest to the clinical genetics community as it highlights considerations in genomic analysis including the importance of careful phenotyping". The thoughtful and helpful comments/suggestions are very much appreciated, and we would like to respond to them as follows:

Major:

1. Study cohort description – would be helpful to include the sequencing modalities applied to this cohort (exome, genome, etc) and sequencers (as there is later discussion of analysis differences due to different platform use).

Response: We appreciate the helpful feedback. We now expand the Methods section to include description of the sequencing modalities applied (exome, genome, panel, optical genome mapping and targeted variant analysis) and NGS platform (HiSeq, Novaseq, Ion Torrent, Ion Proton). This information is now added as a separate column in Table S1.

2. Category 1 i): In the F5711 family description, there is mention of the father's facial dysmorphism but it is not mentioned if this is present or absent in the children (assume absent because not mentioned but better to state what is and is not present given the focus here is phenotype).

Response: Thank you for pointing this out. Yes, the father's facial dysmorphism was considered absent in the children. However, in retrospect, a degree of hypoplastic alae nasi could be appreciated albeit to a much milder degree in the children. Please note that we now replace this family with a newly added family F750 that provides a dramatic example of intrafamilial phenotypic heterogeneity caused by a pathogenic *IRF6* variant. The phenotype ranged widely from frank cleft lip and palate to cryptic lip pits that were not always apparent clinically due to the use of cosmetic fillers. We think this is a better example than the *TBX1* family included in the original submission.

3. Category 1 iv): "Hybrid phenotypes" does not clearly capture what is included in that category in my opinion. Multilocus phenotypes could also suggest a polygenic inheritance rather than monogenic conditions. The words "blended phenotypes", "dual diagnoses" or "multiple genetic diagnoses" could be considered to name that category.

Response: We agree with the respected reviewer that there has not been a consensus to date on how to name this phenomenon. We also agree that the commonly used term "multilocus phenotype" may be misconstrued by some and that "blended phenotype" is a safer alternative. The manuscript and all related files have been thoroughly revised accordingly.

4. Category 2 i) Reporting 1-2 families with interesting variants in a gene not yet associated with disease does not make a novel gene-disease association. The field standard for publishing these is

typically at least 3 cases with extensive phenotypic and molecular description (though case reports are sometimes published typically to raise awareness of a gene to gather more cases to eventually be able to report a novel disease association for a gene). Reporting fewer cases in a table without extensive phenotype data is an insufficient contribution to the literature and these should not be described as novel disease gene assertions.

Response: This is an extremely important point, and we appreciate the opportunity to address it here. The respected reviewer is absolutely correct that the standards set forth by ClinGen for objective and transparent scoring of gene-disease assertions should be followed. Accordingly, the respected reviewer will be pleased to learn that we now fully disclose the gene-disease assertion score in newly added columns M, N and U, V to the revised Table S1. In that table, we show that the overwhelming majority of gene-disease assertions that are novel at the time of analysis (201 out 225, 89.3%) are at least moderate in strength. We would like to emphasize that the main message here is to quantify the contribution of this category to the overall challenge in interpreting clinical exome/genome sequencing. As such, the very small percentage of assertions that are limited in strength will not affect the overall conclusion and are included in full transparency to facilitate future matchmaking.

5. Category 2 iii) “Distinct mutational mechanism” is a subgroup of novel gene-disease assertions (category 2 i)). A name like “novel gene-disease assertion in a gene already known to cause another condition, through a distinct molecular mechanism” or “known gene, novel mutation mechanism” could be considered or if the authors find another way to link it to category 2i.

Response: We have been looking for an efficient way of referring to this category so we very much appreciate the suggested solution by the respected reviewer “known gene, novel mutation mechanism”! The text was edited accordingly.

6. Category 3 i b 6) “There is no clear reason for the discrepancy between the high MAF and observed disease phenotype.” Two variants are given but the allele frequencies are not reported. Checking gnomAD, the *DYNC2H1* variant is in 7 people (https://gnomad.broadinstitute.org/variant/11-102991434-C-T?dataset=gnomad_r2_1) and 1 person in the GME database and the *DALRD3* variant in 1 person (https://gnomad.broadinstitute.org/variant/3-49053669-G-T?dataset=gnomad_r2_1). It would be good to mention the AF in the manuscript and also these AFs are not particularly high.

Response: We regret the confusion that seems to have stemmed from failing to include the allele frequency in the text. These two variants are good examples of how global frequencies can be deceiving. The local allele frequency of the *DYNC2H1* and *DALRD3* variants is 0.001397 and 0.001082, respectively. We must highlight here that “high” allele frequency is defined as >0.001 , which is the cutoff commonly used for autosomal recessive conditions as we clarify in the text. The local allele frequency, already in Table S11, is now included in the text as requested by the respected reviewer.

7. The challenges of non-coding RNA genes should be discussed and probably have its own category. These genes can be captured by exome sequencing, but filtered out and not included in

the variant files for analysis by many pipelines. The example of *RNU4ATAC* disorder is only mentioned in the in silico category. Can the authors comment on diagnoses in non-coding RNA genes in their cohort? Several have been linked to human disorders.

Response: As the respected reviewer is aware, Mendelian diseases caused by lncRNA and other non-coding RNA genes are exceedingly rare. To the best of our knowledge, there has only been one report of a pathogenic lncRNA variant causing a Mendelian phenotype was deletion of a 27 – 63 kb lncRNA locus upstream to the engrailed-1 gene (*EN1*) resulting in congenital limb abnormalities (Allou et al. 2021). The *RNU4ATAC* example is the only example of a non-coding RNA gene we have encountered in our cohort. However, we remain open to the possibility of uncovering more such examples as we employ long-read WGS in cases that remain “negative” despite extensive investigation, and this is currently underway in our lab.

8. International data sharing of variants identified in patients with rare genetic diseases is a major contributor to improve variant interpretation, but also a major challenge. Can the authors discuss this and how data sharing is impacting their analysis?

Response: Another great suggestion! We now expand the penultimate paragraph in Discussion to cover this. Additionally, we now add a new Table S23 that lists novel gene-disease assertions that were corroborated by international collaborations we participated in through data sharing.

Minor comments

1. Line 255 – synonyms instead of synonymous

Response: We apologize for the oversight. This is now corrected.

2. Line 258 – RT-PCR experiments noted altered splicing. Was this variant of interest because of splice predictors? SpliceAI score is 0.35 for donor gain.

Response: This challenging *BCS1L* variant only came to light thanks to autozygome analysis that helped us zero in on this gene and further investigate the seemingly benign yet novel *BCS1L* variant. Interestingly, even SpliceAI failed to predict its effect. Of note, this is now recognized as a founder variant that we identified in at least 2 additional cases, all having had negative WES prior to our discovery, and is included in the newly added Table S22 of founder variants.

3. Line 260 – “a default MAF of <0.001 is often used as a threshold for rare recessive disease” – this is a very conservative threshold compared to what I normally see reported which is <0.01. Is there a reference for this lower threshold?

Response: The ACMG guidelines do not specify a cutoff MAF for their PM2 criterion and instead define it as “extremely low frequency if recessive”. Our lab and many others use 0.001 as a strict cutoff with the understanding that more common pathogenic alleles may be missed and that is indeed the message we are making in this section. In other words, we show that while this strict cutoff is very helpful, one must revisit it when no candidate variants survive this filter. Please note that the original “guide” paper by Bamshad, Nickerson and Shendure (Nature

Reviews Genetics volume 12, pages745–755 (2011)) clearly discussed this issue and their Figure 3 showed that the use of 0.001 MAF as a cutoff for recessive diseases is just as probable of identifying the causal variant as 0.01 MAF if the sample size is large enough.

4. Line 270 – “The variant is only pathogenic when compound heterozygous” – consider adding “with a more damaging variant” as I think that is what you want to convey here.

Response: Excellent suggestion! Text was edited accordingly.

5. Line 364 – should be lowercase pLI

Response: Corrected.

6. Line 368-9 – “Another example is RPGR variant nomenclature, which we found” – add the comma and word which

Response: Edited.

7. Line 423 – capitalize F in Fragile X syndrome

Response: Capitalized.

8. Line 458 – use the term exome rather than whole-exome

Response: Changed.

9. Line 463 – “limited jaw openings to variable severity” to “limited jaw openings of variable severity”

Response: Changed as suggested.

10. Figure 1 bottom left panel categories are unclear from the figure.

Response: Panel was edited in the figure to improve clarity.

11. Figure 5 legend does not have text for parts D and E.

Response: We apologize for the oversight. This is now corrected.

12. In table 2, there is a typo in “mechanism”. Also, “Unusual mutation mechanism” is not clear. A name like “known gene, novel mutation mechanism” or something more descriptive could be considered.

Response: We apologize for the typo, which is now corrected. We appreciate the suggested change, which we now adopt.

13. Table 4 is busy. For ease of reading, the authors could group together cases having the same justification of non-penetrance into categories “inadequate phenotyping”, “true non-penetrance”, etc in the first column.

Response: Excellent suggestion! Table 4 was reorganized accordingly.

14. The abbreviation MAF is not defined in the manuscript though minor allele frequency without abbreviation is listed in the Figure S2 name. The minor allele frequency is the second most common allele in a given population. Given many sites are multiallelic, this is likely not the term the authors mean in all cases and better to use the term “allele frequency”.

Response: Although MAF is commonly used in this context (including in the ACMG guidelines paper), we honor the respected reviewer’s request and change it to “allele frequency” in text and tables.

With many thanks!

Reviewer #2:

We sincerely thank the respected reviewer for the thorough review and thoughtful suggestions, which we believe greatly improved our manuscript. We would like to respond as follows:

1- On the one hand, many of these are challenges that can be encountered and highlighting them could be useful lessons learned. However, there are some major issues with the claims presented that are cause for concern in drawing too many conclusions. I evaluated most of the first 10 or so variants that were called out as causal in the early part of the manuscript, and then a few others here and there and found issues with the majority of them in terms of their qualifications as pathogenic and causal for the phenotype noted. As such, the overall premise of using these cases as exemplars for the challenges noted seems problematic. In general, it does not appear that the authors followed current professional guidelines of evaluating variants for pathogenicity using the 2015 ACMG/AMP guidelines, nor have they ensured that the novel gene-disease associations meet at least moderate by ClinGen standards if they are to consider the findings anything beyond a plausible candidate.

Response: This is an extremely important point and we appreciate the opportunity to address it here. The respected reviewer is absolutely correct in that standards must be followed in claiming causality both at the variant and gene level. Indeed, our lab is a major contributor to the GenCC (Gene Curation Consortium) effort to provide high quality data on gene-disease assertions. Additionally, we always follow the ACMG variant classification guidelines and have even proposed the value of certain features as supporting evidence (e.g. Hum Genet. 2022 Jan;141(1):55-64). We regret not having made this very clear in our methodology. We now made the following changes in response:

- 1) We state the use of ACMG variant classification guidelines and ClinGen gene-disease assertion guidelines in Methods.
- 2) We include ACMG variant classification and, where applicable, ClinGen gene-disease assertion strength in Table S1 (we think redundant inclusion of this information in all other tables will be unnecessary).
- 3) In full transparency, we now state in the manuscript that the majority of the variants identified, are classified as pathogenic or likely pathogenic as per ACMG guidelines. While this leaves a

small 13% of VUS (based on known genes since variants in novel genes must be labeled as VUS according to ACMG guidelines until the gene-disease assertion is published, which was a major incentive for this work), we think there is still great value in including these highly curated variants to facilitate their future reclassification, mostly upgrading given their “hot” nature.

4) Similarly, the section “Genetic Diagnostic Challenges” is now updated to show that the overwhelming majority (201 out of 225, 89.3%) of gene-disease assertions are at least “moderate” in strength. Again, we hope the respected reviewer will agree that the small minority 11% that remain limited at this point are still worth including because this may help matchmaking and subsequent upgrading in the future. Please note that all were indeed included in GeneMatcher. Additionally, they have all been submitted to GenCC for approval of our curation.

2- The respected reviewer raised concerns about the classification of some variants specifically that were highlighted in the text and we appreciate the opportunity to respond as follows:

i) TBX1 - NM_080647.1:c.1158_1159delinsT;p.(Gly387Alafs*73) - This variant is in the terminal exon and therefore not expected to undergo NMD. Insufficient evidence of pathogenicity has been presented, especially given the poor match in phenotype for the family, as the authors point out.

Response: Even if this variant does not trigger NMD, it still qualifies as likely pathogenic because it meets the following criteria: PM2 (absent from controls), PM4 (protein length changes), PP1 (segregation power = 1/8) and PP4 (TBX1-specific phenotype).

INSR - NM_000208.2:c.433C>T;p.(Arg145Cys) – No evidence is provided by authors. There is one zero star entry in ClinVar as Likely Pathogenic but it also has no evidence provided.

Response: This variant is Pathogenic because it satisfies the following criteria: PS3_Strong (functional confirmation, PMID: 22017372 where it was labeled as c.433 C>T (p.R118C)), PP1 (used as strong because we have a segregation power <1/32 across 3 families, founder variant), PP3 (in silico prediction), PP4 (INSR-specific phenotype) and PP5 (ClinVar is a reputable source).

TMPRSS15 - NM_002772.2:c.2808_2809insATCA;p.(Ser937Ilefs*4) – submitted to ClinVar by the authors but with no evidence and no publication link; The PMID listed leads to a paper by the authors but with no evidence presented - this is the most C-terminal pLOF variant in ClinVar and its not clear if it occurs after the NMD site – more evidence for pathogenicity needed.

Response: This variant has indeed been published as an example of a lethal variant in PMID: 34645488 and as such we decided to remove it to keep the revised submission more succinct. Please note that this variant does qualify for Likely Pathogenic because it meets the following criteria: PVS1_Very Strong, PM2_Supporting (LOFTEE score HC).

FSHR - NM_000145.2:c.1030G>A;p.(Val344Met) – unpublished; present in gnomAD but not in ClinVar; no evidence for pathogenicity demonstrated.

Response: We agree with the respected reviewer and have now downgraded this variant due to updated high frequency in our expanded database.

OTX2 - NM_001270525.2:r.98_273del;p.(Pro34Metfs*3) – this variant occurs in a portion of exon 2 that is alternatively spliced out in other transcripts. No other P/LP variants have been observed in this region. Despite gnomAD being highly constrained for pLOF variants (only 1 pLOF), the one present is in this alternately spliced region of exon 2. Further evidence of pathogenicity is needed.

Response: We regret that our quest to keep the text as succinct as possible seems to have limited our ability to include important supporting information. This is an extremely interesting family that we have been studying for many years. The pedigree structure was misleading because we assumed retinitis pigmentosa observed in both branches of the family was genetically homogeneous but this turned out to be false when we found that RP in the other branch was caused by an USH2A splicing variant classified as Pathogenic. The variant is absent in the index and her siblings. This allowed us to zero in on a single critical locus Chr14:51979390-65793170. Interestingly, RNA-seq failed to identify any candidate within this locus. However, upon manually checking the genes, we found two key mouse papers that clearly point to OTX2 as a likely candidate. The first is PMID: 23761884 that showed how OTX2 deficiency can cause retinal dysfunction, which is further supported by the report of OTX2-related RP albeit dominantly (PMID: 19956411). The second is PMID: 21436260 that confirmed infertility among female mice. Thus, we were excited to identify the truncating variant by RT-PCR in OTX2, which we propose to be the first recessive form of OTX2-related RP with infertility. Please note that while this truncating variant does not truncate all isoforms, it still qualifies as Likely Pathogenic based on the following criteria: PM2 (absent from controls, this was checked by tracking a sentinel SNP in the same disease haplotype in the critical locus), PM4 (protein length changes), and PP1 (used as strong because we have a segregation power $<1/32$ in the family).

INSR - NM_000208.2:c.433C>T;p.(Arg145Cys) – No evidence is provided by authors. There is one zero star entry in ClinVar as Likely Pathogenic but it also has no evidence provided.

Response: This variant is Pathogenic because it satisfies the following criteria: PS3 (functional confirmation, PMID: 22017372 where it was labeled as c.433 C>T (p.R118C)), PP1 (used as strong because we have a segregation power $<1/32$ across 3 families, founder variant), PP3 (in silico prediction), PP4 (INSR-specific phenotype) and PP5 (ClinVar is a reputable source).

CD151 - NM_001039490.1:c.493C>T;p.(Arg165*) – there are only 2 P/LP variants reported in ClinVar, spanning at most 4 families so its not surprising that the phenotypic spectrum was poorly defined. Indeed, another paper was just published documenting a third variant in a family with renal agenesis (PMID: 35707593).

Response: We are pleased to see that our claim of phenotypic expansion is supported by this very recent paper, which we now cite. As the respected reviewer noted, phenotypic expansion is most pronounced when dealing with a limited number of reports so while it may not be surprising that we are adding a novel phenotypic aspect, it is still of great utility. Indeed, we had contacted a

renowned group with one of the world's largest CAKUT cohorts about this finding and they found no second hit in their cohort and have even raised doubts this is a real association!

COL25A1 - NM_001256074.2:c.1517delC;p.(Pro506Hisfs*25) and large deletion involving COL25A1 and ZCCHC23 – The COL25A1 gene has been linked to an AR inherited disorder called congenital fibrosis of extraocular muscles. The authors state the gene is linked to a new AD disorder involving fetal akinesia based on two pLOF variants identified. However, no evidence is presented to reconcile the fact that the gene is unconstrained for LOF and littered with pLOF variants throughout the coding region.

Response: As the respected reviewer knows, the LOF constraint metrics available on gnomAD are specifically built for dominant, not recessive genes. Thus, the observation that COL25A1 is unconstrained for LOF in gnomAD has no bearing on the proposed gene-disease assertion. Please note that ClinGen scoring of this assertion is moderate based on the animal model and human genetics data (in addition to our two families, each with a different homozygous truncating variant, we are aware of at least one other group with a similar finding that declined to include their family with ours).

HADHB - NM_000183.2:c.712C>T;p.(Arg238Trp) – This variant was submitted to ClinVar by the authors (as Likely Pathogenic) and by two clinical labs as VUS. Its also present in gnomAD. The authors did not provide any evidence to support pathogenicity.

Response: The variant is classified as Pathogenic as it satisfies the following criteria: PP3_strong (in silico predictions), PM2_Supporting (Low frequency), PP2_Supporting (40 out of 43 non-VUS missense variants in gene *HADHB* are pathogenic), PP5_Supporting (ClinVar contribution), and PP1_Strong (founder variant)

SCLT1 - NM_144643.2:c.778_780del;p.(Glu260del) – The authors state that this case has BBS due to this variant despite the gene being linked to OFD syndrome. However, the gene is not linked to any disorder in OMIM other than a note that says “This variant is classified as a variant of unknown significance because its contribution to orofacioidigital syndrome IX (258865) has not been confirmed” based on a 2014 publication. The genes have been submitted as Limited for BBS and Senior-Loken syndrome in GenCC. So there does not seem to be sufficient evidence for a link to any disorder including BBS and the authors have not presented any data to support their claim.

Response: This is an excellent example of the problem we face with OMIM's slow integration of some gene-disease assertions. We first published SCLT1 as a novel ciliopathy gene in 2014 (PMID 24285566) based on a homozygous LOF variant we identified in a child with typical oral-facial-digital syndrome type IX phenotype, the classical ciliopathy phenotype in knockout mice (polycystic kidneys) and the known essential role of SCLT1 in ciliogenesis. Subsequently, we identified the same SCLT1 variant in in a child with a BBS phenotype comprising intellectual disability, obesity, coloboma, renal failure (s/p renal transplant), retinopathy, midline cleft, epilepsy, atopy, panhypopituitarism, and undescended left testicle (PMID: 27894351). Finally, the same founder variant was subsequently identified by us in two sisters with classical Bardet-Biedl syndrome (PMID: 32055034). We have emailed OMIM with all these updates years ago

and are still waiting for this to reflect in the database. Remarkably, several other groups have subsequently identified SCLT1-related ciliopathies (see Bardet–Biedl syndrome in two unrelated patients with identical compound heterozygous SCLT1 mutations PMID: 32253632, SCLT1-related disease as a rare cause of cone dystrophy with subtle systemic associations resembling ciliopathy <https://iovs.arvojournals.org/article.aspx?articleid=2782010>, Compound heterozygous splice site variants in the SCLT1 gene highlight an additional candidate locus for Senior-Løken syndrome PMID: 30425282, Bardet-Biedl Syndrome Caused by Skipping of SCLT1 Complicated by Microvesicular Steatohepatitis PMID: 33132306, Biallelic loss-of-function alleles of the SCLT1 gene cause a variable phenotypic spectrum of retinal ciliopathies <https://iovs.arvojournals.org/article.aspx?articleid=2742842>). In view of this background, we hope the respected reviewer will see the value of adding this case with *SCLT1*-related BBS. Now this gene is classified as strong based on GenCC criteria. The variant is a VUS, however, the deleted amino acid is the last codon at the exon-intron boundary.

CDK10 - NM_052988.3:c.139delG;p.(Glu47Argfs*21) – this one does appear to be pathogenic but it is listed as unpublished but was published by another group in 2017 (PMID: 28886341), is present in OMIM and submitted to ClinVar.

Response: There seems to be some confusion. The case itself is not published, but the variant is as already indicated in Table S1 (the columns J, R, X, and AF refer to whether the variant is novel while column AG refers to whether the case is published).

PIK3C2G - NM_004570.5:c.3101G>A;p.(Arg1034His) - This is the first example described for hybrid phenotypes and was found in a patient with a MERTK deletion causing RP. The authors argue that the PIK3C2G variant is responsible for their insulin resistance. However, no evidence has been presented for why this variant was assumed causal. The gene has not been implicated in any diseases in OMIM. A search in ClinVar finds one entry from a research lab proposing it as a candidate gene for Tracheoesophageal fistula and a literature search finds one small non-replicated GWAS study from 15 years ago that finds an association with 2 SNPs in this gene and provides an OR of 2. I do not think this is sufficient evidence to say that this gene or variant has any role in the patient's insulin resistance.

Response: We sincerely apologize for this oversight. We have specifically discussed this variant and decided to drop it, but it did make it to the final version inadvertently. This example is now removed.

3- The authors state “The clinical utility of causal variant identification in Mendelian diseases entails the provision of a precise diagnostic label, informing management decisions and empowering individuals (patients and unaffected carriers) to make reproductive choices.” I would add “understand disease risk in family members and future generations” as an equally important element of clinical utility.

Response: added as suggested.

4- The authors state “When the underlying molecular diagnosis was considered challenging due to factors unique to a particular family but not others, only that family counted towards the tally

of challenging cases. Otherwise, all families that were susceptible to a given challenge were counted.” This is confusing as I’m trying to figure out why they can’t just state “All families that were susceptible to a given challenge were counted.” Is there something about the first sentence that makes the second sentence on its own inaccurate?

Response: Not at all! We now adopt the respected reviewer’s suggested language.

5- The authors include a method for OPT but it seems most cases were analyzed by WES. Please include all approaches in the methods.

Response: The reason we only described the methodology for OGM is that we have already published our pipeline for exome, panel and autozygosity mapping. In response to the respected reviewer’s request, we now add a new Supplemental Methods that describes each of the methods.

6- Please include a better description of the cohort including where cases were recruited from. Were they all research subjects or were some cases from clinical testing services? What was the race/ethnicity/ancestry of the population? Was it enriched for consanguinity?

Response: We now expand the description of the cohort to include the requested details i.e. research vs clinical testing, ancestry, and consanguinity.

7- For category 1- Phenotype-related: iii) Allelism, how much evidence is there for these new allelic diseases at the variant and gene level, especially the 41 that are not even entered in OMIM, which has a low bar of evidence for entry. For example, how many of the distinct allelic disorders had sufficient evidence for the gene-disease relationship to reach at least moderate evidence according to ClinGen standards? It would be useful to delineate how many valid new allelic diseases were due to distinct inheritance patterns (e.g. AR vs AD) and/or mutation types (e.g. pLOF vs GOF/missense only).

Response: As we discussed above, we do not think OMIM necessarily has a low bar of evidence for entry (see the SCLT1 example). In fact, we are perplexed by the apparent resistance of OMIM sometimes to consider allelic disorders despite ample evidence. Perhaps it is appropriate to cite just one more example in this regard. THSD1-related non-immune hydrops fetalis is now by far the most common autosomal recessive form of NIHF in our country caused by a single variant. The gene-disease assertion is based on numerous families published by us and others (see PMID: 26036949, PMID: 28749478, PMID: 28600779, PMID: 34645488, PMID: 33569873, PMID: 30055085 and <https://www.degruyter.com/document/doi/10.1515/crpm-2021-0030/html?lang=de>) including LOF variant, as well as the compelling animal model for THSD1 role in vascular permeability (see PMID: 27895300). Despite repeated appeals, we are told by OMIM that this is still being deliberated internally. This is why we now rely more on GenCC to more efficiently disseminate our findings to the wider community with objective and transparent scoring. In response to the respected reviewer’s comment, we now include our scoring as submitted to GenCC for all these allelic disorders (Table S1). As for the mechanism i.e. distinct inheritance pattern vs. mutation type, this information is now displayed in Table S4 column J.

8- For the Gene-related section 2- i) on novel gene-disease assertions, the list spans genes which have been subsequently implicated in disease as the authors state, but it also includes genes that are simply candidates based on a single family (e.g. DNAJC17), and even 23 genes that are obscured with a “a” symbol as the authors do not wish to reveal the discovery. Similar “a” annotations are in other tables. Please remove these genes if you are unwilling to identify them. Also, it would be important to distinguish those genes that do not yet have sufficient evidence to be validly implicated in disease and are still candidates. Use of ClinGen’s framework for gene disease validity is suggested. It would also be useful to know how many are in the GenCC database. It seems the senior member of this paper is a submitter to GenCC yet these genes have not yet been submitted. Is there a reason why?

Response: As mentioned above, we now fully share our objective scoring of the gene-disease assertions as submitted to GenCC. We are fully committed to data sharing and have been a major contributor to the GenCC effort. The only genes we obscured with a “a” symbol are genes that we do not have the final say on because they are part of collaborations. These collaborators do not wish to have these genes revealed at this stage and we have to honor their preference. However, we fully share all the data related to these genes below for the eyes of the respected reviewer. We believe deleting these cases will unfairly skew the percentages of the respective data metrics so we wish to retain them. Please note that the sentence “we highlight 120 gene-disease assertions that are not yet listed in OMIM (Table S7)” already excludes the masked genes denoted as “a” in the table.

Pedigree	Gene	Variant	Zygosity	GenCC Classification	Evidence
F4110	APOBEC2	NM_006789.3:c.190G>A;p.(Gly64Arg)	Homozygous	Moderate	Variant evidence (1); Segregation evidence (2); Animal model (4); Gene-Level Experimental Evidence (2)
F4196	OLA1	NM_013341.4:c.427C>T;p.(Arg143*)	Homozygous	Moderate	Variant evidence (2); Segregation evidence (3); Gene-Level Experimental Evidence (4);

					Functional validation on going
F4576	ADAMTS9	NM_182920.2:c.860G>A;p.(Arg287His)	Homozygous	Moderate	Variant evidence (1); Segregation evidence (0.5); Animal Model (4); Functional validation on going
F5098	MAST3	NM_015016:c.3575C>T;p.(Pro1192Leu)	Homozygous	Moderate	Variant evidence (1); Segregation evidence: (1); Animal model (4); Gene-Level Experimental Evidence (2)
F4631 and F5936	EXOC1	NM_001024924.1:c.1342C>T;p.(Arg448*) and NM_001024924.1:c.1955G>A;p.(Ser652Asn)	Homozygous	Moderate	Variant evidence (2); Segregation evidence (4.5); Gene-Level Experimental Evidence (2); Animal model (2)
F6116	PARP16	NM_017851.4:c.275C>A;p.(Ser92*)	Homozygous	Moderate	Variant evidence (2); Segregation evidence (4.5); Functional validation under progress

F6468	AP3D1	NM_001261826.3:c.98C>T;p.(Ala33Val)	Homozygous	Moderate	Variant evidence (1); Segregation evidence (0.5); Functional validation on going; Two additional families with different variants
F6578	VPS39	NM_015289.2:c.86T>C;p.(Leu29Pro)	Homozygous	Moderate	Variant evidence (1); Segregation evidence (3); Animal Model (4); Functional validation on going
F681	VPS25	NM_032353.3:c.22C>T;p.(Pro8Ser)	Homozygous	Moderate	Variant evidence (1); Segregation evidence (1.5); Animal Model (4); Functional validation on going
F7683	CSF3	NM_001178147.2:c.415C>T;p.(Gln139*)	Homozygous	Moderate	Variant evidence (2); Segregation evidence (1.5); Animal Model (4); Functional validation on going
F8356	DMAP1	NM_019100:c.511T>C;p.(Phe171Leu)	Homozygous	Strong	Variant evidence

					(1); Segregation evidence (1.5); Animal Model (4); Working on manuscript with multiple variants (Collaborative paper)
F5644 and F8441	PSKH1	NM_006742.3:c.361C>T;p.(Arg121Trp)	Homozygous	Moderate	Variant evidence (2); Segregation evidence (4); Animal model (4); Undergoing validation (three families)
F8576	AARSD1	NM_001261434.2:c.560delG;p.(Gly187Valfs*28)	Homozygous	Moderate	Variant evidence (2); Segregation evidence (1.5); Animal model (4); Undergoing validation
F9094	FADS6	NM_001720.5:c.913G>A;p.(Glu305Lys)	Homozygous	Limited	Variant evidence (1); Segregation evidence (0.5); Animal model (4)
F9104	WDR27	NM_001202550.1:c.1622+4G>C	Homozygous	Limited	Variant evidence (1); Segregation evidence (0.5);

					Animal model (4)
F5867	RBM47	NM_001098634.2:c.1623delC:p.(Tyr541*)	Homozygous	Moderate	Variant evidence (2); Segregation evidence (4); Animal model (4); Undergoing validation
F6140	WDR59	NM_030581.3:c.2887G>A:p.(Gly963Arg)	Homozygous	Moderate	Variant evidence (1); Segregation evidence (4); Animal model (4); Undergoing validation (three families with founder variant)
F6350	CREBRF	NM_153607.3:c.475delT:p.(Ser159Hisfs*57)	Heterozygous (De novo)	Moderate	Variant evidence (4); Segregation evidence (3); Undergoing validation/data collection (two families)
F8571	MRPL49	NM_004927.4:c.263G>A;p.(Arg88His)	Homozygous	Moderate	Variant evidence (1); Segregation evidence (4); Undergoing validation
F8843	MORF4L	NM_206839.2:c.608T>C:p.(Leu203Pro)	Homozygous	Limited	Variant evidence (1);

					Segregation evidence (1.5); Animal model (2)
F9165, F9263, and F9304	DBR1	NM_016216.3:c.200A>G;p.(Tyr67Cys)	Homozygous	Moderate	Variant evidence (1); Segregation evidence (2); Animal model (4); Undergoing validation and founder mutation
F8316	NCLN	NM_020170.4:c.185-1del;r.185del;p.(Gly62Alafs*7)	Homozygous	Moderate	Variant evidence (2); Segregation evidence (0.5); Animal model (4); Undergoing validation
F4418	EFNB2	NM_004093:c.830G>A;p.(Arg277His)	Homozygous	Limited	Variant evidence (1); Segregation evidence (0.5)
F3942	C3ORF67	NM_198463:c.508C>T;p.(Arg170*)	Homozygous	Strong	Variant evidence (2); Segregation evidence (4); Animal model (4); Undergoing validation
F7764	MRPL47	NM_020409.2:c.646C>T;p.(Arg216*)	Homozygous	Moderate	Variant evidence (2); Segregation evidence (0.5);

					Additional variants from another group
F8895	SLAIN2	NM_020846.2:c.52C>T;p.(Gln18*)	Homozygous	Limited	Variant evidence (2); Segregation evidence (0.5)
F8600	ECRG4	NM_032411.3:c.79+2T>C	Homozygous	Limited	Variant evidence (2); Segregation evidence (0.5)
F8845	ADGRA2	NM_032777.10:c.1737delC;p.(Asn579fs*109)	Homozygous	Moderate	Variant evidence (5); Segregation evidence (0.5); Animal model (4)
F5571	VPS52	NM_001289176.1:c.619G>A;p.(Glu207Lys)	Homozygous	Limited	Variant evidence (1); Segregation evidence (0.5)
F6667	PDIA6	NM_005742:c.898delG;p.(Ala300fs*22)	Homozygous	Moderate	Variant evidence (2); Segregation evidence (0.5); Animal model (4)
F4263	HMGXB3	NM_014983.3:c.3600C>G;p.(Tyr1200*)	Homozygous	Moderate	Variant evidence (2); Segregation evidence: (5); Animal model (4);

					Undergoing validation
F8274	NUBP2	NM_012225.4:c.334G>A;p.(Ala112Thr)	Homozygous	Moderate	Variant evidence (1); Segregation evidence (0.5); Animal model (4); Additional family from Matchmaking
F6391	PSTK	NM_001363531.2:c.166G>C;p.(Asp56His)	Homozygous	Limited	Variant evidence (1); Segregation evidence (0.5)
F9685	C1orf109	NM_001350767.2:c.224G>C;p.(Arg75Pro)	Homozygous	Limited	Variant evidence (1); Segregation evidence (0.5)
F4476	MAP4K4	NM_145687.4:c.3469C>T;p.(Arg1157*)	Heterozygous (De novo)	Moderate	Variant evidence (2); Segregation evidence (0.5); Animal model (4); Undergoing validation
F4138	CCDC142	NM_032779.4:c.1668_1676del;p.(Gln558_Leu560del)	Homozygous	Limited	Variant evidence (1); Segregation evidence (0.5)
F6350	IRAK1	NM_001569.4:c.446G>A;p.(Gly149Asp)	Hemizygous	Limited	Variant evidence (1); Segregation (0.5)

F8913	PAPSS1	NM_005443.5:c.1774C>T;p.(Arg592Cys)	Homozygous	Moderate	Variant evidence (1); Segregation evidence (0.5); Animal model (4); Undergoing validation
F9552, F7104, F9593, F9596, and F9599	PATJ	NM_001350145.3:c.780del;p.(Phe260Leufs*3)	Homozygous	Strong	Variant evidence (4); Segregation evidence (4); Undergoing validation and total 12 families are recruited for the same gene
F9586	AXIN1	NM_181050.3:c.2167C>T;p.(Arg723*)	Homozygous	Moderate	Variant evidence (2); Segregation evidence (1.5); Gene-Level Experimental Evidence (2); Undergoing validation
F33 and F9171	MTDH	NM_178812.4:c.1718_1721del;p.(Lys573Argfs*33) and NM_178812.4:c.862delG;p.(Glu288Lysfs*4)	Homozygous	Moderate	Variant evidence (4); Segregation evidence: (4); Animal model (1); Undergoing validation (two different variants)

					from two families)
F8064	COPS9	NM_138336.1:c.157-2A>G	Homozygous	Limited	Variant evidence (2); Segregation evidence: (0.5)
F3612	FLVCR1	NM_014053.4:c.1593+5_1593+8del	Homozygous	Moderate	Variant evidence (2); Segregation evidence (2); Animal model (4); Gene-Level Experimental Evidence (2)
F6584	USP39	NM_001256728:c.440G>A;p.Arg147Gln	Homozygous	Moderate	Variant evidence (1); Segregation evidence (2.5); Animal model (4)
F7448	HMCN1	NM_031935.3:c.12690+1->TAAGT	Homozygous	Moderate	Variant evidence (2); Segregation evidence (0.5); Animal model (4); Undergoing validation
F2492	BIRC3	ENST00000263464.3:c.1763G>A;p.(Cys588Tyr)	Homozygous	Moderate	Variant evidence (1); Segregation evidence (2); Animal model (4)
F3422	WDR47	NM_001142551.2:c.578G>A;p.(Arg193His)	Homozygous	Strong	Variant evidence (4);

					Segregation evidence (4); Gene-Level Experimental Evidence (4)
--	--	--	--	--	--

9- In section 3-i-a, the authors describe all splice region variants as TDV (transcript deleterious variants) even before they have been evaluated. I would suggest describing them as splice region variants and only identify the subset as TDV once proven to impact splicing. Also, please be sure all variants are being evaluated and reported use the MANE Select or MANE Plus Clinical transcripts as the current standard.

Response: The term TDV has already been peer-reviewed and accepted (PMID: 32552793). In that paper we describe the impact of those TDV at the RNA level in full details. Given that a very small percentage of these TDV have not been verified on RTPCR, we agree with the respected reviewer that we should have a more conservative approach to the name, and we now use the term “tentative TDV” instead. We appreciate the great suggestion to use the MANE Select or MANE Plus Clinical transcripts. This is now clearly mentioned in Methods. Reassuringly, the overwhelming majority of our variants are already listed using MANE Select or MANE Plus Clinical so we only had to update a very small number of variants in the relevant tables.

10- In section 3-i-b, the authors highlight variants that were challenging due to high MAF. Yet the majority of the variants in the list are well-established pathogenic variants listed as such in ClinVar. Was a standard filter not used to identify pathogenic variants in ClinVar? Other variants were called out as not yet being recognized as more common rare diseases (e.g. WASHC5 and NT5DC1) yet evidence for pathogenicity was not presented as noted below and each was only seen in one proband (RECQL4 also seen in only one). Please provide evidence for pathogenicity and a common role in disease.

WASHC5 - NM_014846.3:c.2849A>G;p.(Lys950Arg) - No evidence for pathogenicity provided. VUS in ClinVar

NT5DC1 NM_152729.3:c.1114T>C;p.(Ser372Pro) -- No evidence for pathogenicity provided.

CEP250 - NM_007186.2:c.1826C>T;p.(Ala609Val) – No evidence for pathogenicity provided. VUS in ClinVar

Response: We appreciate the opportunity to clarify this important issue. Of course, we are not suggesting that the sickle cell disease variant for example is challenging per se (this is already mentioned in the text). What we are trying to convey is that the commonly used MAF cutoff of 0.001 runs the risk of filtering out disease-causing variants. When a disease is well established and already known to be common as with the sickle cell disease example, this should pose no problem as pointed out by the respected reviewer. The problem lies in those that are not yet well

established. For example, when we first identified the founder variants in *C21orf57*, *CYP2U1* and *ADAT3*, we were dismissive because their MAF was perceived to be too high. It was only after we and others established these founder variants are the most common autosomal recessive causes of intellectual disability/developmental delay in Saudi Arabia that our pipelines started retaining them despite their MAF. In fact, we are very excited to share yet another illustrative variant that we encountered during revision and is now included. In a patient of mine with ADHD, his parents reported occasional “blisters” but they were always gone by the time I saw him in clinic. I requested clinical exome, which revealed a VUS in *TGM5*:NM_201631.3:c.1335G>C;p.(Lys445Asn), which the clinical lab quickly retracted because they reclassified it as “benign” because of its high MAF. However, upon further investigation, this turned out to be an ancient founder variant that was first reported in Tunisia (see PMID: 19440220) in a patient with the same phenotype. Upon discussing this with a colleague dermatologist (now part of the Mendeliome Group), she told me that she has another patient with the same phenotype and when we looked up the exome for that patient we identified the same founder variant. Thus, we now have a compelling founder variant that results in a relatively mild phenotype, which explains why it has not come to surface yet. Please note that we do have a skin biopsy from the latter patient that shows the classical cleft between the stratum corneum and stratum granulosum (see expanded Figure S3). We agree with the respected reviewer that our bar for including variants with high MAF should be high. Indeed, with the exception of CEP250 variant (which we now remove), all the variants in that table qualify as at least likely pathogenic. As for the specific examples raised by the respected reviewer, below is their classification:

WASHC5 - NM_014846.3:c.2849A>G;p.(Lys950Arg) - The variant is classified as Likely Pathogenic because it meets the following criteria: PP3_Strong (in silico predictions), PM2_Supporting (absent in gnomAD), PP1_Strong (segregation power), and PB1_Supporting (28.5% of VUS missense variants are benign)

NT5DC1 NM_152729.3:c.1114T>C;p.(Ser372Pro) - The variant is classified as Likely Pathogenic because it satisfies the following criteria: PM2_Supporting (absent in gnomAD), PP3 Supporting (in silico predictions), and PP1_Strong (segregation power)

CEP250 - NM_007186.2:c.1826C>T;p.(Ala609Val) - we agree with the respected reviewer and now this variant is removed.

11- In the “5- Positional mapping-related limitations” section, the authors describe confusion over the apparent familial segregation discordance related to ROH due to assuming IBD when the ROH was due to IBS. Yet its not clear if they ever looked at the specific variant under question in these situations, which I presume would have shown the accurate segregation with disease if it was causal. While I understand that use of ROH in consanguineous families can be a useful way to focus in on the most likely regions implicated in disease, I would assume that it is only a first pass strategy and not something relied on.

Response: As the respected reviewer will appreciate, we will not have identified the causal variant had we not actually looked beyond what ROH analysis had initially suggested. The

message we are conveying here is exactly what the respected reviewer stated i.e. positional mapping is a very helpful strategy but has its limitations and as such should not be viewed as foolproof. Please note that in our experience with 158 families where the phenotype was mapped to a single locus, the causal variant was always found within the locus. We think the problem with ROH analysis is typically when no such level of “fine mapping” is possible.

12- The authors state “Contrary to the common belief that the missing diagnostic yield of exome is mostly related to technical limitations...”. However, I’m not aware that this is a common belief and instead I think most believe the missing yield is due to our inability to interpret most variation as causal. If the authors wish to state this, please reference sources.

Response: It is very common in the literature on the diagnostic yield of exome sequencing to indicate that this may be due to inherent limitation of exome and that whole genome may be able to address that. Indeed, it is a common practice among clinical geneticists to request genome sequencing on cases with negative exome results and high index of suspicion for a genetic etiology, and this is in fact a standard procedure at UDN (see PMID: 32730690). This is based on a number of studies showing a higher yield of genome vs. exome (see PMID: 30002876 for an excellent meta-analysis). We now cite an excellent preprint by some giants in this field (<https://arxiv.org/abs/2301.07363>).

13- The authors state “... since it makes the difference between counseling them for a 50% recurrence risk on the assumption of non-penetrance to a nearly 0% recurrence risk on the assumption of recessiveness and non-carrier spouse.” In most settings of genetic disease counseling, recurrence risk is often focused on the chance of having a second child with the disease, which in this case would be 25%, not 0%. It may be worth adding that point as well.

Response: There seems to be some confusion and we appreciate the opportunity to clarify. When a clinical geneticist counsels a carrier individual with an autosomal recessive variant and non-carrier spouse, the risk of an affected child is negligible. Perhaps the confusion stems from the term “recurrence” so we now replace it with “risk of an affected child” so we are not referring to “recurrence” of something that has already happened in the family.

14- The authors note that they “... provide an interpretation framework for variants in 52 genes that we note harbor homozygous loss of function class variants with no apparent consequences phenotypically” yet I don’t see this list anywhere in the materials provided. Also, have the pLOF variants been evaluated to ensure they are subject to LOF as opposed to escaping NMD or removing the variant’s effect through alternative splicing, etc?

Response: These 52 genes and their variants are indeed included in Table S14 that we submitted. Perhaps this table was not made available by the editorial office to the respected reviewer? In response to the respected reviewer’s concern, we now update this table to include detailed analysis of these pLOF variants and show that they all have high LOFTEE scores except for seven cases that have low confidence scores and 3 cases where the score was undetermined.

REVIEWER COMMENTS

Reviewer #1 (Remarks to the Author):

AlAbdi et al have adequately responded to my comments. Overall, I continue to think this article will be of interest to the clinical genetics community though it remains a very dense “listing” manuscript to read. There is a lot of great data in the supplementary tables.

For Category 3 i b 6), I do not think “high AF” is an accurate term. Another option would be: “However, variants with higher allele frequency accounted for the molecular diagnosis in 253 families”. Or AF above cut-off (or filter).

And are the variants in this category in ClinVar as pathogenic or likely pathogenic, because previously classified variants should be considered even if the AF is >0.001.

Consider abbreviating allele frequency as AF to save space. While I recommended not using MAF, it’s fine to use AF.

DECIPHER should be capitalized when mentioned in the manuscript (page 18 but may be elsewhere).

The large supplemental tables are better shared as excel files than pdfs.

Reviewer #2 (Remarks to the Author):

I have reviewed the author’s responses to my feedback and appreciate the added detail that has now been provided. Overall, I retain some concern about incorrect classification of many variants. As some general feedback, PM2 should not be applied at moderate strength per ClinGen guidelines. The authors apply PP4 even when cases are not a match for the phenotype associated with the gene let alone is high specificity as defined for this evidence code. The authors apply PVS1 when variants are not predicted to undergo NMD. The authors apply PP5 which is not appropriate to apply to ClinVar submission and has been removed from the ACMG standards by ClinGen. Furthermore, my overall concern is underscored by the number of variants that were removed by the authors after my first review as well as the category of “g- Distraction by other variants” where they note that in 231 families (15.2%) the search for the causal variant was derailed by another variant. The fact that this is a frequent occurrence

emphasizes that variants are being routinely misinterpreted. That said, it is beyond the scope of the review process to debate the classification of hundreds of variants. And I appreciate that the authors have now listed the ACMG codes applied which allows more transparency for other groups to re-review the data and come to their own conclusions. The main point of the paper is to highlight ways in which genome analysis can be challenging or go awry during the interpretation process and I think the authors have made that point very clear. Indeed, it is common for labs to misinterpret variants in cases so hopefully these lessons may better educate labs on these challenges. They have also shared detailed case level data which is important to add to the evidence base for genes and variants. As such, I feel this manuscript will be a useful contribution to the field of rare disease genomic analysis. I have a few more minor points that came up during the second review that are listed below. In addition, please be sure that all variants are submitted to ClinVar with supporting evidence and connection to any published manuscripts and any new gene-disease relationships are submitted to GenCC.

In the section labeled “3-Variant-related, i) b-High allele frequency” I would not consider these variants as having high allele frequencies given that your threshold is quite low and many recessive alleles are much more common than this. Instead, I would refer to them as “Population-specific founder variants”. In the following paragraph, instead of saying “1-The variant causes a common disease...” I would instead state “1-The variant is a founder variant...” given that it is definitely not what one would consider a common disease nor even a more frequent rare disease.

The authors note finding ALDOB:NM_000035.4:c.448G>C;p.(Ala150Pro) in a patient with cancer. This finding seems to be an incidental finding that did not contribute to a blended phenotype as no phenotype had been shared regarding the hereditary fructose intolerance until after the genetic result was returned. I would either exclude this case or put it in a section related to incidental findings.

In the sentence beginning “e- Multivariant alleles: When more than one pathogenic variant is detected in the same gene...” I would replace the word ‘gene’ with ‘allele’.

Reviewer #1:

AlAbdi et al have adequately responded to my comments. Overall, I continue to think this article will be of interest to the clinical genetics community though it remains a very dense “listing” manuscript to read. There is a lot of great data in the supplementary tables.

We are pleased to learn that the respected reviewer assessed that we have adequately addressed their concerns. The kind comments that “I continue to think this article will be of interest to the clinical genetics community” and “There is a lot of great data in the supplementary tables” are greatly appreciated. We also appreciate the helpful comments and suggestions that we would like to address as follows:

1- For Category 3 i b 6), I do not think “high AF” is an accurate term. Another option would be: “However, variants with higher allele frequency accounted for the molecular diagnosis in 253 families”. Or AF above cut-off (or filter).

We agree and have now used “AF above cut-off” to avoid confusion.

2- And are the variants in this category in ClinVar as pathogenic or likely pathogenic, because previously classified variants should be considered even if the AF is >0.001 .

Excellent point and one we appreciate the opportunity to clarify here. As the respected reviewer knows, the designation by ClinVar that a variant is pathogenic or likely pathogenic is far from being the final say and this is well established in the literature (see PMID: 29625023, PMID: 33108757, and PMID: 27884173). In fact, some “pathogenic” variants in ClinVar have been downgraded specifically because of population frequency. Of course, we are not suggesting that very well established disease-causing variant such as the sickle cell disease variant are challenging because their AF is above the traditional cutoff. Rather, we are simply drawing the community’s attention to the overall contribution of variants whose AF is above the traditional cutoff to the disease-causing variants pool and how these can be challenging.

3- Consider abbreviating allele frequency as AF to save space. While I recommended not using MAF, it’s fine to use AF.

Allele frequency is now changed to AF as suggested.

4- DECIPHER should be capitalized when mentioned in the manuscript (page 18 but may be elsewhere).

Capitalized as suggested.

5- The large supplemental tables are better shared as excel files than pdfs.

Absolutely! Unfortunately, however, the journal’s submission website forces us to upload these files as PDF. We are more than happy to upload them as Excel file if that can be made possible and it will actually make our submission much easier.

With many thanks!

Reviewer #2:

I have reviewed the author's responses to my feedback and appreciate the added detail that has now been provided.

We are indebted to the respected reviewer for their continued attempts to improve the quality of this manuscript and the kind comment that "this manuscript will be a useful contribution to the field of rare disease genomic analysis". We appreciate the opportunity to take advantage of the additional helpful suggestions to make further improvements as follows:

1- Overall, I retain some concern about incorrect classification of many variants. As some general feedback, PM2 should not be applied at moderate strength per ClinGen guidelines.

The respected reviewer will be pleased to learn that we have indeed applied PM2 as a supporting evidence as per the updated ClinGen guidelines. The only exception that seems to have misled the respected reviewer was TBX1 - NM_080647.1:c.1158_1159delinsT;p.(Gly387Alafs*73), which remained likely pathogenic even after we corrected the use of PM2 from moderate to supporting.

2- The authors apply PP4 even when cases are not a match for the phenotype associated with the gene let alone is high specificity as defined for this evidence code.

The respected reviewer is correct in that our use of PP4 was at times not very accurate. We have now carefully reviewed our five entries for the application of PP4. Fortunately, only 2 variants had inappropriate use of the PP4 criteria and only one variant was downgraded from likely pathogenic to VUS as a result.

3- The authors apply PVS1 when variants are not predicted to undergo NMD.

As the respected reviewer knows, the updated guidelines on PVS1 (PMID: 30192042) do not limit its application to NMD and extend it to instances where the truncated protein segment is known to be critical for the protein function. Furthermore, if the truncated portion is >10% of the protein length PVS1 can be used with the strength "Strong". As such and after careful review of our variants, none of the variants were downgraded to VUS or below after adjusting the strength of the PVS1 criteria.

4- The authors apply PP5 which is not appropriate to apply to ClinVar submission and has been removed from the ACMG standards by ClinGen.

The respected reviewer is correct in that a recent communication indicated that PP5 use based on ClinVar entries is problematic (PMID: 29543229). We have now carefully revised every single use of PP5 (469 entries) and removed it when it was based on ClinVar but retained it if it was based on a high quality publication. Fortunately, this led to the downgrading of only 7 variants from likely pathogenic to VUS.

5- Furthermore, my overall concern is underscored by the number of variants that were removed by the authors after my first review as well as the category of "g- Distraction by other variants" where they note that in 231 families (15.2%) the search for the causal variant was derailed by

another variant. The fact that this is a frequent occurrence emphasizes that variants are being routinely misinterpreted. That said, it is beyond the scope of the review process to debate the classification of hundreds of variants. And I appreciate that the authors have now listed the ACMG codes applied which allows more transparency for other groups to re-review the data and come to their own conclusions. The main point of the paper is to highlight ways in which genome analysis can be challenging or go awry during the interpretation process and I think the authors have made that point very clear. Indeed, it is common for labs to misinterpret variants in cases so hopefully these lessons may better educate labs on these challenges. They have also shared detailed case level data which is important to add to the evidence base for genes and variants. As such, I feel this manuscript will be a useful contribution to the field of rare disease genomic analysis. I have a few more minor points that came up during the second review that are listed below.

We fully agree with the respected reviewer that the main message of the manuscript is to highlight ways in which genome analysis can be challenging or go awry during the interpretation process. The changes made during revision are indeed a reflection of how dynamic this process is and we have opted to share it exactly as it is in full transparency. As the respected reviewer correctly noted, the full disclosure of how variants were classified allows for a fully transparent assessment of the best evidence we have at the moment and we appreciate the opportunity to have these included in the relevant manuscript files.

6- In addition, please be sure that all variants are submitted to ClinVar with supporting evidence and connection to any published manuscripts and any new gene-disease relationships are submitted to GenCC.

We now include the ClinVar submission ID for our variants. Please note that all new gene-disease relationships have been submitted to GenCC as we indicated in the previous submission.

7- In the section labeled “3-Variant-related, i) b-High allele frequency” I would not consider these variants as having high allele frequencies given that your threshold is quite low and many recessive alleles are much more common than this. Instead, I would refer to them as “Population-specific founder variants”. In the following paragraph, instead of saying “1-The variant causes a common disease...” I would instead state “1-The variant is a founder variant...” given that it is definitely not what one would consider a common disease nor even a more frequent rare disease.

The respected reviewer is absolutely correct in that referring to some genetic diseases as “common” may cause confusion. We now change “3-Variant-related, i) b-High allele frequency” to “3-Variant-related, i) b- Allele frequency above cut-off” as suggested by Reviewer #1. We also change “1-The variant causes a common disease...” to “1-The variant is a founder variant that causes a disease with a previously unrecognized relatively high incidence...”.

8- The authors note finding ALDOB:NM_000035.4:c.448G>C;p.(Ala150Pro) in a patient with cancer. This finding seems to be an incidental finding that did not contribute to a blended phenotype as no phenotype had been shared regarding the hereditary fructose intolerance until after the genetic result was returned. I would either exclude this case or put it in a section related to incidental findings.

We regret the apparent misunderstanding here. As we indicate in the paper, the index individual presented for a strong family history of cancer but was also found upon reviewing the medical history to have severe sweet aversion. Thus, the ALDOB finding is not an incidental finding since it fully explains one of the two phenotypes identified in the initial presentation. We did not suggest anywhere that the ALDOB finding has anything to do with the strong family history of cancer because this would be wrong as pointed out by the respected reviewer.

9- In the sentence beginning “e- Multivariant alleles: When more than one pathogenic variant is detected in the same gene...” I would replace the word ‘gene’ with ‘allele’.

Changed as suggested.

With many thanks!

REVIEWERS' COMMENTS

Reviewer #2 (Remarks to the Author):

The authors have adequately addressed my remaining comments. I have no further points of feedback.